# ⬙ THE FIRST DROP OF INK:
# Nonlinear Impact of Misleading Information in Long-Context Reasoning

**Muhan Gao** [1]   **Zih-Ching Chen** [2]   **Kuan-Hao Huang** [1]

## Abstract

As large language models are increasingly deployed in retrieval-augmented generation and agentic systems that accumulate extensive context, understanding how distracting information affects long-context performance becomes critical. Prior work shows that semantically relevant yet misleading documents degrade performance, but the quantitative relationship between the proportion of distractors and performance remains unstudied. In this work, we systematically vary the hard-distractor proportion in fixed-length contexts, revealing a striking nonlinear pattern: as the proportion of hard distractors increases, performance drops sharply within the first small fraction, while the remainder of the range yields only marginal additional decline. We term this " THE FIRST DROP OF INK" effect, analogous to how a single drop of ink contaminates water. Our theoretical and empirical analyses grounded in attention mechanics show that hard distractors capture disproportionate attention even at small proportions, with diminishing marginal impact as their proportion grows. Controlled experiments further show that filtering gains mainly come from context-length reduction rather than distractor removal; substantial recovery requires reducing the hard-distractor proportion to near zero, highlighting the importance of upstream retrieval precision.

## 1. Introduction

Recent advances in long-context language models (Anthropic, 2025) have given rise to applications that aggregate

[1]Department of Computer Science and Engineering, Texas A&M University, College Station, TX, USA [2]NVIDIA AI Technology Center, NVIDIA Corporation, Santa Clara, CA, USA. Correspondence to: Muhan Gao <muhangao@tamu.edu>, Kuan-Hao Huang <khhuang@tamu.edu>.

*Proceedings of the $43^{rd}$ International Conference on Machine Learning*, Seoul, South Korea. PMLR 306, 2026. Copyright 2026 by the author(s).

extensive documents into a single context. Deep research pipelines (OpenAI, 2025), for instance, autonomously retrieve and synthesize information from numerous sources, while long-document analysis systems enable users to query entire books or legal documents (Ke et al., 2026; Chang et al., 2024; Guha et al., 2023; Su et al., 2025). These applications accumulate large volumes of text, often exceeding 100K tokens, before generating a final response. However, as models ingest more documents, they inevitably encounter information that is topically relevant yet ultimately misleading.

Prior work on long-context language models has primarily focused on how the position (Liu et al., 2024) and length (Bianchi et al., 2025; Levy et al., 2025; 2024) of relevant information affect performance. Less attention has been paid to the surrounding context itself. While research on short-context reasoning tasks (Shi et al., 2023; Yang et al., 2025a) and retrieval-augmented generation (RAG) systems (Lee et al., 2026; Jin et al., 2025) demonstrates that distractors can cause non-negligible performance drops, and Hong et al. (2025) reveals that this effect amplifies as context length grows, how performance degrades in long contexts as the proportion of misleading documents increases remains unexplored. A natural question arises: *how does performance change as the proportion of distractors grows in long-context reasoning?*

In this work, we systematically vary the proportion of hard distractors within fixed-length contexts and identify THE FIRST DROP OF INK effect as in Figure 1: as hard distractor proportion increases, performance drops sharply within the first small fraction, then plateaus with only marginal further decline. We provide a theoretical analysis grounded in the softmax attention mechanism, showing that attention on the gold document is a convex function of hard distractor proportion, with empirical validation on retrieval heads (Wu et al., 2025; Zhang et al., 2025b). This explains the observed nonlinearity: hard distractors dominate the softmax denominator even at small proportions, implying that partially removing them yields negligible recovery and only near-complete removal restores performance.

These findings challenge the prevailing assumption in long-

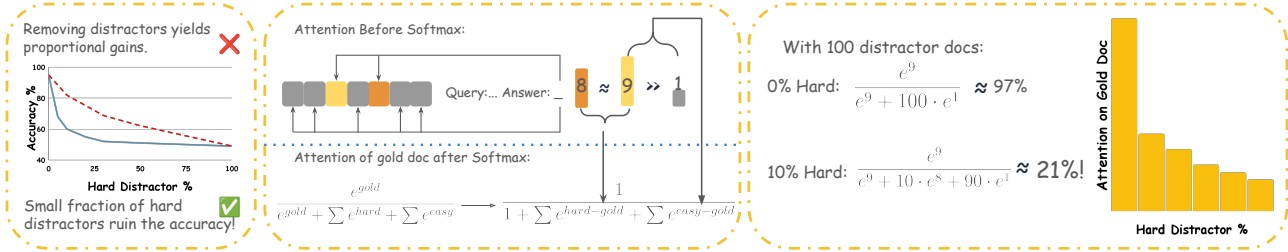

*Figure 1.* THE FIRST DROP OF INK effect. **Left:** Conventional linear assumption (top, red dashed line) versus empirically observed nonlinear degradation (bottom, blue curve): a small fraction of hard distractors is sufficient to severely degrade accuracy. **Middle:** Hard distractors receive similar attention logits as gold documents ($8 \approx 9 \gg 1$), dominating the softmax competition even at low proportions. **Right:** With 100 distractor documents, attention on gold drops 76% by adding only 10% hard distractors. This convex relationship explains THE FIRST DROP OF INK.

context applications that accumulating more documents improves performance. As long as even a small fraction of hard distractors remains in the context, performance is severely degraded; consequently, post-hoc filtering in most cases yields only marginal recovery. This suggests that preventing hard distractors from entering the context in the first place is more critical than filtering them afterward.

**Contribution.** (1) We identify THE FIRST DROP OF INK effect across multiple models and datasets: as the proportion of hard distractors increases, performance degrades sharply within the first small fraction, then plateaus. (2) We provide a theoretical explanation showing that attention on the gold document is a strictly convex function of hard distractor proportion, and validate this empirically through attention logit measurements on retrieval heads. (3) We design controlled experiments to disentangle the effects of context length and distractor composition, showing that conventional filtering yields gains primarily from context reduction, and removing hard distractors only provides substantial benefit when their proportion is reduced to near zero.

## 2. Related Work

**Long-context understanding and evaluation.** The ability to process long context has emerged as a critical capability for large language models (LLMs), with the context window extended from 4K to over 1M tokens (Anthropic, 2025; Team, 2024; Xiao et al., 2024; Ding et al., 2024; Peng et al., 2024). This expansion has motivated efforts to more effectively understand and evaluate long-context ability.

Among various evaluation approaches, the "Needle-in-a-Haystack" (NIAH) paradigm (Kamradt, 2023) is preferred due to its controllability and ease of construction (Hsieh et al., 2024b; Yen et al., 2025; Bai et al., 2024; Zhang et al., 2024a). A "needle" (a fact or short passage required to answer a query) is inserted into a "haystack" of unrelated filler text, and the model must locate and use the needle while ignoring the surrounding context.

Under this paradigm, Liu et al. (2024) identify the "Lost-in-the-Middle" phenomenon: LLMs prioritize information at the start and end of contexts while neglecting middle portions. This limitation persists across various scenarios (Lee et al., 2025; Gao et al., 2024), motivating follow-up studies to develop methods for mitigating positional bias (Hsieh et al., 2024a; Zhang et al., 2024b; Wang et al., 2025). Beyond position, needle length also affects retrieval accuracy (Bianchi et al., 2025; Levy et al., 2025).

Recent work has also examined the haystack itself. In original settings (Kamradt, 2023; Hsieh et al., 2024b), the haystack consists of irrelevant documents, posing no semantic confusion with the target needle. Yang et al. (2025b) use synthetically generated biographies to improve coherence between needles and haystack, better approximating realistic retrieval conditions. Further studies introduce semantically related distractors into the haystack and observe non-negligible performance degradation (Lee et al., 2026; Hong et al., 2025). However, under the long context setting, how performance varies with the proportion of distractors in the haystack remains unexplored.

**Information aggregation in agentic systems.** The rise of agentic AI has fundamentally transformed how LLMs interact with external information. Rather than responding to a single query with a fixed context, modern systems such as deep research pipelines (OpenAI, 2025; Zhang et al., 2025a), multi-agent collaboration frameworks (Wu et al., 2023; Hong et al., 2024; Li et al., 2023), and tool-augmented agents (Schick et al., 2023; Qin et al., 2024; Yao et al., 2023) autonomously gather, aggregate, and synthesize information across multiple retrieval rounds. These systems routinely accumulate contexts exceeding 100K tokens before producing a final response (Singh et al., 2025). Recent work has shown that even context length alone can degrade performance (Du et al., 2025), further underscoring the challenges of information aggregation at scale.

This information aggregation process introduces an unavoidable challenge: while gathering relevant information, these systems inevitably accumulate unhelpful or misleading documents along the way (Jin et al., 2025; Shi et al., 2023;

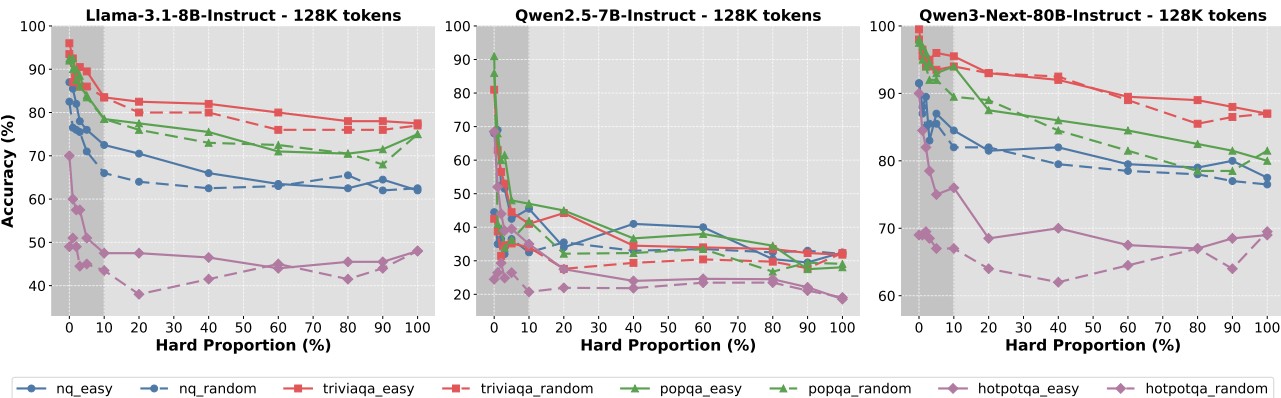

*Figure 2.* Accuracy as a function of hard distractor proportion at 128K context length across three models (`Llama-3.1-8B-Instruct`, `Qwen2.5-7B-Instruct`, and `Qwen3-Next-80B-Instruct`) on Natural Questions, TriviaQA, PopQA and HotpotQA. Across all configurations, introducing the first 10% of hard distractors (shaded region) causes steep performance degradation, while further increases yield only marginal decline. Despite substantial variation in absolute accuracy across datasets (e.g., HotpotQA shows the lowest baseline due to multi-hop reasoning), the nonlinear pattern persists, illustrating the THE FIRST DROP OF INK effect.

Yang et al., 2025a). Prior work demonstrates that such noisy retrieval can significantly degrade LLM performance (Cuconasu et al., 2024; Yoran et al., 2024). In response, filtering and reranking have become standard techniques for improving RAG performance, operating under the assumption that removing distractors yields substantial gains (Glass et al., 2022; Yoran et al., 2024). However, these findings are derived from relatively short contexts of only a few thousand tokens. Whether the same assumptions hold, and whether existing mitigation strategies remain effective, as context windows scale to 128K tokens and beyond, remains underexplored.

## 3. Nonlinearity in Distractor Effects

We study how the proportion of hard distractors affects model performance in a multi-document question answering setting, where a language model must locate relevant information among retrieved passages. Formally, given a query $q$, a gold passage $\mathcal{J}^*$ containing the answer, and a set of $N$ distractor passages $\{\mathcal{P}_1, \ldots, \mathcal{P}_N\}$, the model must attend to $\mathcal{J}^*$ to produce the correct answer. We categorize distractors into three types based on their semantic relevance to $q$: easy ($\mathcal{E}$), random ($\mathcal{R}$), and hard ($\mathcal{H}$), and systematically vary their proportions to study the resulting performance degradation. We begin by describing our experimental setup (§3.1) and then present our findings (§3.2).

### 3.1. Experimental Setup

**Dataset.** We use Natural Questions (Kwiatkowski et al., 2019), TriviaQA (Joshi et al., 2017), PopQA (Mallen et al., 2023), and HotpotQA (Yang et al., 2018), covering both single-hop and multi-hop reasoning. Each sample is a tuple $(q, a, \mathcal{J}^*)$, where $q$ is the question, $a$ is the gold answer, and $\mathcal{J}^*$ is the gold passage from which $a$ can be derived.

**Distractors.** To control distractor difficulty, we use three categories of passages with varying degrees of relevance to the query $q$: (1) **Easy** ($\mathcal{E}$): repetitions of a single filler sentence *"The grass is green. The sky is blue. The sun is yellow..."*; (2) **Random** ($\mathcal{R}$): arbitrary passages sampled from the Wikipedia `2019-08-01` dump from KILT (Petroni et al., 2021); (3) **Hard** ($\mathcal{H}$): semantically related passages retrieved from Wikipedia using BM25, which are topically relevant to $q$ but do not contain the answer. We use `gpt-4o-mini` to examine each hard distractor and filter out those that contain the answer in any form (including paraphrases or alternative expressions of $a$, prompt in §C), ensuring that hard distractors are genuinely misleading rather than inadvertently providing correct information. All three categories of distractors are normalized to approximately 100–150 tokens to avoid length bias.

**Input.** Given a target context length $T$ and a hard distractor proportion $p \in [0, 1]$ (see §A for specific values), we construct the input by concatenating the gold passage $\mathcal{J}^*$ with distractors sampled to fill the context. For each dataset, we consider two mixing strategies: (1) *easy-hard mixing*, where proportion $p$ of distractors are from $\mathcal{H}$ and $(1-p)$ are from $\mathcal{E}$ (e.g., `nq_easy`), and (2) *random-hard mixing*, where proportion $p$ of distractors are from $\mathcal{H}$ and $(1-p)$ are from $\mathcal{R}$ (e.g., `nq_random`). All passages are randomly shuffled before concatenation to avoid positional bias. For each setting (dataset $\times$ context length $\times$ hard proportion), we sample 200 examples for evaluation.

**Evaluation.** Hsieh et al. (2024b) employs string containment matching to evaluate QA tasks:

$$\text{Accuracy} = \frac{1}{N} \sum_{i=1}^{N} \max_{r \in R_i} \mathbb{K}[\texttt{lower}(r) \subseteq \texttt{lower}(p_i)]$$

where $p_i$ denotes the model prediction, $R_i$ is the set of

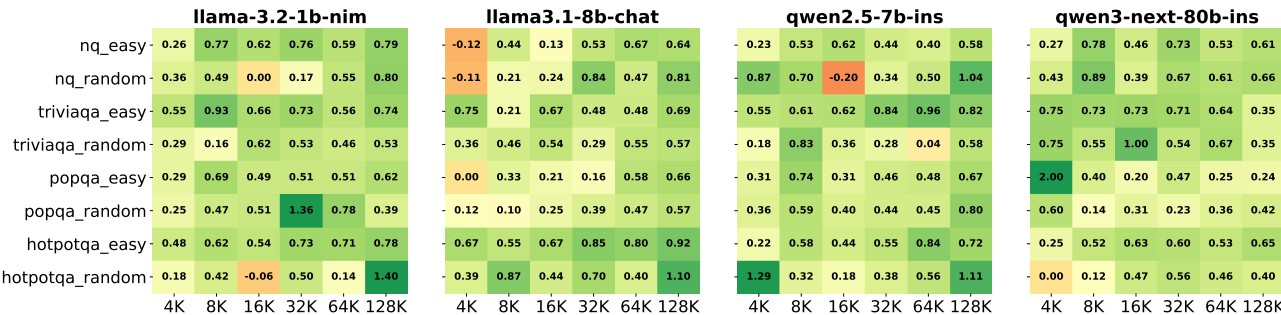

*Figure 3.* Drop ratio of accuracy degradation across different context lengths, models, and datasets. The drop ratio measures the fraction of total performance loss that occurs in the first 10% of hard distractors. A linear degradation would yield **0.1**. Negative values indicate the first 10% of hard distractors does not further degrade performance. Darker green indicates higher drop ratios (stronger nonlinearity), while orange/red indicates values near or below the linear baseline. The prevalence of green across the table confirms that the THE FIRST DROP OF INK effect is consistent across models and datasets.

reference answers, and $\mathbb{1}[\cdot]$ is the indicator function. A prediction is considered correct if any reference answer appears as a substring within it (case-insensitive). However, we observe that string matching suffers from false negatives (e.g., "3" vs. "three", "Bill Clinton" vs. "William Jefferson Clinton", "1986-2013" vs. "from 1986 to 2013"). Taking HotpotQA as an example, we identify 36 out of 200 samples (18%) where the model's response is semantically correct but marked incorrect by string matching.

Therefore, we follow Yen et al. (2025) and use `gpt-4o-mini` as an LLM judge to verify correctness. The judge receives only the gold document $\mathcal{J}^*$, question $q$, correct answer $a$, and model output, and determines whether the output is semantically correct (prompt in §C). We manually check 100 samples on each dataset and find the judge produces only 17 false negatives in total (4.25%), consistent with prior findings that LLM judges achieve Cohen's $\kappa$ of 0.72–0.91 with human judgment (Yen et al., 2025).

### 3.2. THE FIRST DROP OF INK Effect

We demonstrate the results for 3 models on the length of 128K tokens in Figure 2, more detailed results for all the models can be found in §A.

**Accuracy shows a nonlinear relationship with hard distractor proportion.** As shown in Figure 2, the initial increase in hard distractor proportion (0–10%, shaded region) causes disproportionately large performance drops compared to subsequent increases (10–100%). To quantify this asymmetry, we compute the ratio of accuracy drop in the 0–10% region versus the total drop from 0–100% in Figure 3:

$$\text{Drop Ratio} = \frac{\text{Acc}(0\%) - \text{Acc}(10\%)}{\text{Acc}(0\%) - \text{Acc}(100\%)}$$

A linear degradation would yield a ratio of 0.1; significantly higher values indicate *front-loaded* degradation. For example, on `nq_easy` at 128K context, `Qwen2.5-7B-Instruct` exhibits a drop ratio of 0.58,

which means 58% of the total degradation occurs in the first 10% of hard distractors. These results show the THE FIRST DROP OF INK effect, which is contrary to the linear degradation assumption where each additional hard distractor contributes equally and the expected ratio would be 0.1.

## 4. The First Drop Matters Most

In this section we theoretically analyze the mechanistic reason of THE FIRST DROP OF INK effect based on the transformer's attention mechanism.

### 4.1. Preliminaries and Notations

**Attention mechanism.** For a sequence of $T$ tokens with hidden representations $\{h_i\}_{i=1}^T \in \mathbb{R}^d$, the attention mechanism computes query, key, and value projections:

$$q_i = W_Q h_i, \quad k_j = W_K h_j, \quad v_j = W_V h_j$$

The attention logits, weights, and output are:

$$z_{i,j} = \frac{q_i^\top k_j}{\sqrt{d_k}}, \ \alpha_{i,j} = \frac{\exp(z_{i,j})}{\sum_{\ell=1}^T \exp(z_{i,\ell})}, \ o_i = \sum_{j=1}^T \alpha_{i,j} v_j \quad (1)$$

An autoregressive model predicts the next token based on the last position's attention over all preceding tokens.

**Retrieval task.** When predicting the answer to query $q$, the relevant information lies in the gold passage $\mathcal{J}^*$, a span of tokens within the context. Retrieval succeeds if the model attends sufficiently to $\mathcal{J}^*$ when generating the answer. Prior work on retrieval heads (Wu et al., 2025; Zhang et al., 2025b) has shown that the attention weight $\alpha_{i,\mathcal{J}^*} := \sum_{j \in \mathcal{J}^*} \alpha_{i,j}$ on the target passage strongly correlates with downstream accuracy: higher attention mass on the gold passage leads to higher probability of correct answer generation.

**Logit margin.** Let $i$ denote the position of the last token, from which the model generates the answer. For a passage $\mathcal{P}$ spanning multiple tokens, we define its aggregate logit as

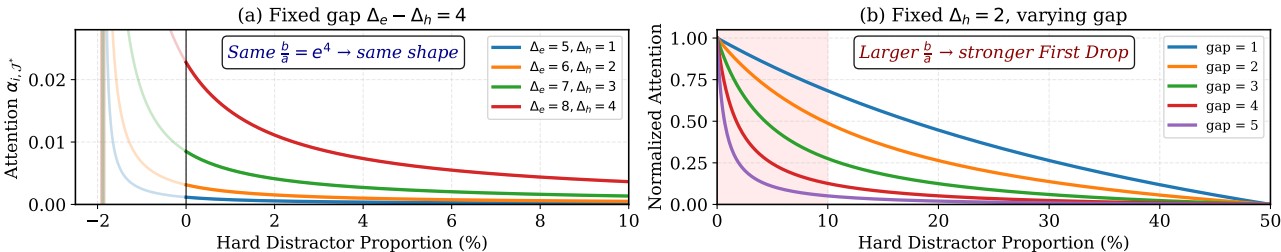

*Figure 4.* Two controlling factors of the theoretical attention curve (Remark 4.3). (a) When the margin gap $\Delta_e - \Delta_h$ is fixed at 4, all curves share identical shape (same $b/a = e^4$) but differ in vertical position, controlled by $1/a$. Faded lines extend into $p < 0$ to illustrate the shape equivalence. (b) When $\Delta_h$ is fixed at 2, increasing $\Delta_e$ enlarges the ratio $b/a$, producing more convex curves and amplifying the "First Drop of Ink" effect (shaded region, 0–10%).

$z_{i,\mathcal{P}} := \frac{1}{|\mathcal{P}|} \sum_{j \in \mathcal{P}} z_{i,j}$, representing how strongly the last token attends to passage $\mathcal{P}$. The margin between the target passage $\mathcal{J}^*$ and a distractor passage $\mathcal{P}$ is:

$$\Delta_{\mathcal{P}} := z_{i,\mathcal{J}^*} - z_{i,\mathcal{P}}$$

where $z_{i,j}$ is the attention logit defined in Eq. (1).

In our two mixing strategies (§3.1), we always mix hard distractors ($\mathcal{H}$) with a weaker distractor type (either easy or random). For notational simplicity in the following analysis, we use $\mathcal{E}$ to denote the **weaker** distractor set and $\Delta_e$ to denote its characteristic margin:

$$\Delta_e := \frac{1}{|\mathcal{E}|} \sum_{\mathcal{P} \in \mathcal{E}} \Delta_{\mathcal{P}}, \quad \Delta_h := \frac{1}{|\mathcal{H}|} \sum_{\mathcal{P} \in \mathcal{H}} \Delta_{\mathcal{P}}$$

Since hard distractors are semantically more similar to the query and compete more strongly for attention, we have $\Delta_h \ll \Delta_e$. **We empirically validate this in §5.**

### 4.2. Theoretical Explanation of Nonlinear Degradation

**Lemma 4.1** (Attention Weight with Mixed Distractors). *Consider a context of total length $T$ tokens, consisting of: (1) A target passage $\mathcal{J}^*$ with $T_g$ tokens; (2) Distractor passages with $T_d$ tokens, where proportion $p \in [0, 1]$ are from $\mathcal{H}$ and $(1 - p)$ are from the weaker category; (3) Other tokens (query, instructions) with $T_o$ tokens, where $T = T_g + T_d + T_o$. The aggregate attention weight on the target passage is:*

$$\alpha_{i,\mathcal{J}^*}(p) = \frac{1}{1 + (1 - p) \cdot a + p \cdot b + c}$$

*where $a := T_d \cdot e^{-\Delta_e}$ and $b := T_d \cdot e^{-\Delta_h}$ represent the aggregate contributions from weaker and hard distractors respectively, and $c := T_o \cdot e^{-\Delta_o}$ denotes the contribution from other tokens.*

*Proof.* From Eq. (1), the aggregate attention weight on the target passage is:

$$\alpha_{i,\mathcal{J}^*} = \frac{\sum_{j \in \mathcal{J}^*} \exp(z_{i,j})}{\sum_{j=1}^{T} \exp(z_{i,j})}$$

The denominator decomposes as:

$$\underbrace{\sum_{j \in \mathcal{J}^*} \exp(z_{i,j})}_{\text{target passage } \mathcal{J}^*} + \underbrace{\sum_{j \in \mathcal{E}} \exp(z_{i,j})}_{\text{weaker distractors } \mathcal{E}} + \underbrace{\sum_{j \in \mathcal{H}} \exp(z_{i,j})}_{\text{hard distractors } \mathcal{H}} + \underbrace{\sum_{j \in \mathcal{O}} \exp(z_{i,j})}_{\text{other tokens } \mathcal{O}}$$

By the definition of logit margin, for tokens in weaker distractors we have $z_{i,j} = z_{i,\mathcal{J}^*} - \Delta_e$, and for tokens in hard distractors we have $z_{i,j} = z_{i,\mathcal{J}^*} - \Delta_h$. Thus:

$$\sum_{j \in \mathcal{E}} \exp(z_{i,j}) = (1 - p) \cdot T_d \cdot \exp(z_{i,\mathcal{J}^*}) \cdot e^{-\Delta_e}$$

$$\sum_{j \in \mathcal{H}} \exp(z_{i,j}) = p \cdot T_d \cdot \exp(z_{i,\mathcal{J}^*}) \cdot e^{-\Delta_h}$$

$$\sum_{j \in \mathcal{O}} \exp(z_{i,j}) = T_o \cdot \exp(z_{i,\mathcal{J}^*}) \cdot e^{-\Delta_o}$$

Substituting and factoring out $\exp(z_{i,\mathcal{J}^*})$ from numerator and denominator, and noting that $T_g \ll T_d$ (the target passage is small relative to distractors):
$\alpha_{i,\mathcal{J}^*}(p)$

$$= \frac{1}{1 + (1 - p) \cdot T_d \cdot e^{-\Delta_e} + p \cdot T_d \cdot e^{-\Delta_h} + T_o \cdot e^{-\Delta_o}}$$

$$= \frac{1}{1 + (1 - p)a + pb + c}$$

where $a := T_d \cdot e^{-\Delta_e}$, $b := T_d \cdot e^{-\Delta_h}$, and $c := T_o \cdot e^{-\Delta_o}$. □

**Lemma 4.2** (Monotonicity and Convexity). *Let $f(p) = \alpha_{i,\mathcal{J}^*}(p) = \frac{1}{1+(1-p)a+pb+c}$.*

*Then $f'(p) < 0$ (strictly decreasing) and $f''(p) > 0$ (strictly convex) for all $p \in [0, 1]$.*

*Proof.* Let $D(p) := 1 + (1-p)a + pb + c = 1 + a + c + p(b - a)$. Since $\Delta_h \ll \Delta_e$ (hard distractors have smaller margins), we have $e^{-\Delta_h} > e^{-\Delta_e}$, and thus $b = T_d \cdot e^{-\Delta_h} > T_d \cdot e^{-\Delta_e} = a$. Let $\gamma := b - a > 0$. Then $D(p) = 1 + b + c + p\gamma$.

First derivative:

$$f'(p) = -\frac{\gamma}{D(p)^2}$$

Since $\gamma > 0$ and $D(p) > 0$, we have $f'(p) < 0$.

Second derivative:

$$f''(p) = \frac{2\gamma^2}{D(p)^3}$$

Since $\gamma^2 > 0$ and $D(p) > 0$, we have $f''(p) > 0$. $\qquad\square$

*Remark* 4.3 (Simplified Form for Large Context). When $a, b \gg 1$ (i.e., $T_d$ is sufficiently large), the constant term in the denominator becomes negligible:

$$\alpha(p) = \frac{1}{1 + (1-p)a + pb} \approx \frac{1}{(1-p)a + pb}$$
$$= \frac{1}{a} \cdot \frac{1}{1 + p\left(\frac{b}{a} - 1\right)}$$

This reveals two key insights, as illustrated by Figure 4:

- **Vertical position** is controlled by $1/a = e^{\Delta_e}/T_d$: larger $\Delta_e$ shifts the curve upward.

- **Curve shape** is controlled solely by $b/a = e^{\Delta_e - \Delta_h}$: only the margin gap matters, not the absolute values.

> **Takeaway:** The monotonicity confirms that increasing the hard distractor proportion always hurts attention on the target. The strict convexity further implies that this degradation is *front-loaded*: the first few percent of hard distractors cause disproportionately large drops, while subsequent increases have diminishing impact. Together, these properties provide the theoretical basis for THE FIRST DROP OF INK effect.

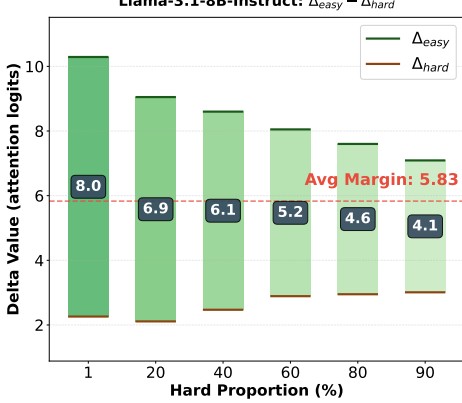

*Figure 5.* Empirical measurement of logit margins $\Delta_e$ and $\Delta_h$ on retrieval heads for Llama-3.1-8B-Instruct. Green bars show $\Delta_e$ (margin to easy distractors) and brown bars show $\Delta_h$ (margin to hard distractors). The gap $\Delta_e - \Delta_h$ (annotated values) remains substantial across all hard proportions, with an average of 5.83. This validates the theoretical assumption $\Delta_h \ll \Delta_e$.

## 5. Validation of Theoretical Explanation

One natural question is: does the model truly exhibit a clear gap between attention on semantically similar versus dissimilar distractors (i.e., $\Delta_h \ll \Delta_e$)? To answer this, we measure $\Delta_e$ and $\Delta_h$ by computing the attention logit difference between the target passage and distractor passages. Rather than averaging across all attention heads or selecting a specific layer, we follow Zhang et al. (2025b) to identify the sparse subset of heads (approximately 1–2%) responsible for retrieving relevant information from context, as their attention mass directly correlates with retrieval success.

Specifically, given a query $q$ and context containing gold passage $\mathcal{J}^*$ among distractors, we score each attention head $h$ by the attention mass it allocates from query tokens to the gold passage. While Zhang et al. (2025b) use post-softmax attention weights, we observe numerical underflow in long contexts (128K tokens) where attention weights become vanishingly small. We therefore use pre-softmax logits instead:

$$\text{Score}_h(q) = \frac{1}{|q|} \sum_{t_q \in q} \frac{1}{|\mathcal{J}^*|} \sum_{t_d \in \mathcal{J}^*} z_h^{t_q \to t_d}$$

where $z_h^{t_q \to t_d}$ is the attention logit from query token $t_q$ to document token $t_d$ in head $h$. For each setting of dataset and hard proportion, we use 50 samples to identify the top-scoring heads as retrieval heads, and then measure $\Delta_e$ and $\Delta_h$ on the remaining 150 samples. The identified heads are highly stable: Pearson correlation between train and test scores on the top-16 heads is $0.96 \pm 0.01$, and Spearman rank correlation across all heads is $0.99 \pm 0.00$. We report results for `Llama-3.1-8B-Instruct` in Figure 5; results for `Llama-3.2-1B-Instruct` and additional details are provided in §B.

**Margin separation confirms theoretical assumption.** Figure 5 shows the measured margins for `Llama-3.1-8B-Instruct` across different hard proportions. We observe a clear and consistent separation: $\Delta_e \approx 7$–$10$ while $\Delta_h \approx 2$–$3$, yielding an average gap of 5.83. This confirms our theoretical assumption that $\Delta_h \ll \Delta_e$. To understand the practical implication, consider a 128K context with $T_d = 128000$ distractor tokens. The ratio $b/a = e^{\Delta_e - \Delta_h} \approx e^{5.83} \approx 340$ means that each hard distractor token contributes $340\times$ more to the softmax denominator than an easy distractor token. Even at just 10% hard proportion, hard distractors account for $\frac{0.1 \times 340}{0.1 \times 340 + 0.9 \times 1} \approx 97\%$ of the total distractor contribution, completely dominating the attention competition.

**Shrinking gap reinforces THE FIRST DROP OF INK effect.** One might argue that the margin gap $(\Delta_e - \Delta_h)$ decreases as hard proportion increases: from 8.0 at 1% to 4.1 at 90%, and wonder whether this undermines our theory. In fact, the opposite is true: this observation reinforces THE

FIRST DROP OF INK effect. The largest margin gap occurs precisely when the hard proportion is lowest, meaning the first few hard distractors enjoy the maximum competitive advantage ($b/a \approx e^{8.0} \approx 2980$) over easy distractors. As more hard distractors are added, the gap shrinks and so does their marginal impact ($b/a \approx e^{4.1} \approx 60$ at 90%). This is exactly the pattern our theory predicts: first drops sharply and then plateaus.

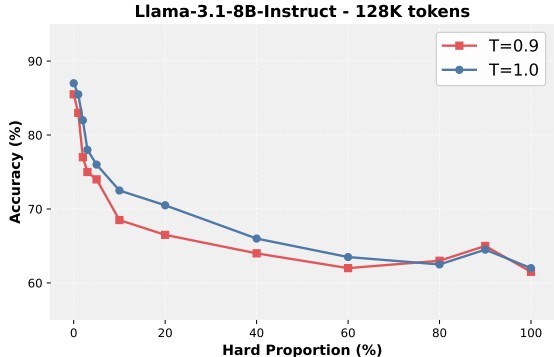

*Figure 6.* Effect of softmax temperature scaling on accuracy across hard proportions (nq_easy, Llama-3.1-8B-Instruct). Lower temperature ($\tau = 0.9$) consistently degrades performance despite theoretically sharpening attention toward the target, indicating that inference-time temperature adjustments cannot mitigate THE FIRST DROP OF INK effect.

## 6. Implications for Mitigation Strategies

### 6.1. Inference-Time Temperature Scaling

Our theoretical analysis in §4 indicates that the softmax function's exponential nature causes hard distractors to dominate the attention competition despite their lower logits than the target. A natural hypothesis is that decreasing the softmax temperature $\tau$ during inference could "sharpen" the attention distribution, amplifying the target passage's advantage as the highest-logit tokens (Figure 7). Specifically, we modify the attention computation as:

$$\alpha_{i,j} = \frac{\exp(z_{i,j}/\tau)}{\sum_{\ell=1}^{N} \exp(z_{i,\ell}/\tau)}$$

where $\tau < 1$ produces a sharper distribution that concentrates more attention on the target passage.

**Results.** Figure 6 shows the effect of temperature scaling on Llama-3.1-8B-Instruct across different hard proportions on nq_easy. Contrary to our hypothesis, decreasing temperature consistently *degrades* performance across all hard proportions.

**Why does this fail?** Although lower temperature theoretically sharpens attention toward the target, the model was trained with $\tau = 1$ and its learned dynamics are calibrated to this setting. Modifying $\tau$ at inference time disrupts these

dynamics, degrading performance even when the attention distribution appears more favorable. Indeed, effective temperature scaling typically requires adjustment during training or fine-tuning (Ryan, 2024; Ram et al., 2025), as the model must learn to adapt its representations to the modified softmax behavior. Our results suggest that THE FIRST DROP OF INK effect cannot be mitigated through simple inference-time interventions.

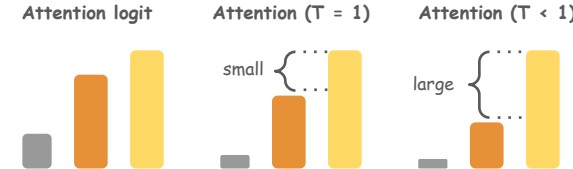

*Figure 7.* Effect of temperature scaling on attention distribution. From left to right: pre-softmax logits, attention weights at $\tau = 1$, and attention weights at $\tau < 1$. Colors denote easy distractors, hard distractors, and target passage. Lower temperature sharpens the softmax, suppressing hard distractors while maintaining attention on the target.

### 6.2. Incremental Filtering of Hard Distractors

Our main experiments vary the hard proportion while fixing the context length. In practice, however, filtering removes unwanted passages entirely, reducing the overall context length. This creates a confound: when filtering improves performance, is the gain due to removing hard distractors, or simply due to shorter context? We design two experiments to disentangle these factors: (1) We compare **Filter Hard versus Filter Random** with symmetric starting compositions, removing only hard or weaker distractors respectively, isolating the effect of filtering strategy. (2) We perform **Proportional Reduction**, shrinking context length while holding the hard distractor ratio fixed by removing both types proportionally, isolating the pure effect of context length. Both experiments are conducted on Llama-3.1-8B-Instruct and Qwen2.5-7B-Instruct for all four datasets.

**Filter Hard vs. Filter Random.** **Filter Hard** begins with 80% hard distractors ($\approx 102K$) and 20% random distractors ($\approx 26K$), progressively removing hard distractors and reducing context length by approximately 20K tokens at each step until 27K tokens remain. **Filter Random** begins with the reversed composition (20% hard, 80% random) and removes random distractors at the same pace. Both experiments end at 27K tokens but with opposite compositions: Filter Hard ends with nearly all random distractors, while Filter Random ends with nearly all hard distractors. Table 1 summarizes the composition at each step.

Figure 8 shows the results. From 131K to 47K tokens, both filtering strategies yield nearly identical performance gains regardless of whether hard or random distractors are

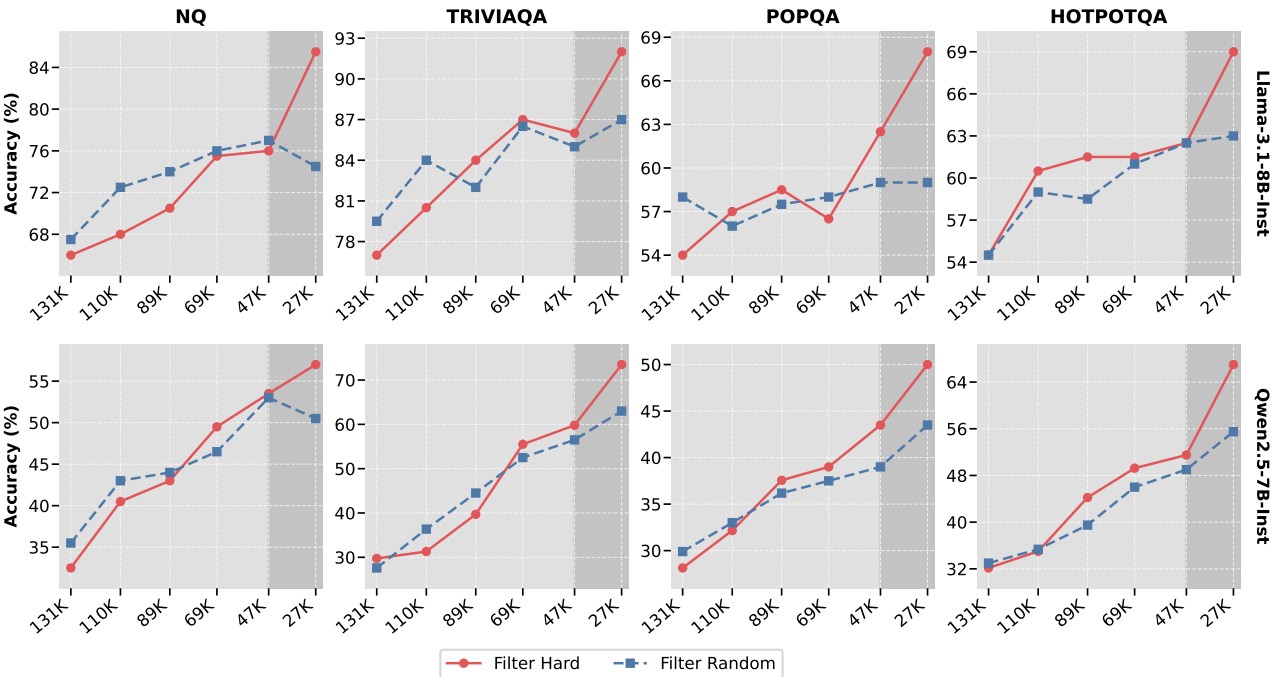

*Figure 8.* **Filter Hard vs. Filter Random.** Both strategies yield similar gains from removing the first 80K tokens, indicating that performance recovery comes from context length reduction rather than filtering strategy. The two strategies begin to diverge below 47K tokens (shaded region), where Filter Hard has a near-zero hard distractor proportion. This suggests that the gains from partial filtering are largely attributable to context reduction rather than the removal of hard distractors themselves.

removed. This indicates that the performance improvement has little to do with the filtering strategy itself, and comes almost entirely from reducing context length. However, the two curves diverge between 47K and 27K tokens (shaded area). At 27K, Filter Hard has reduced the hard proportion to near zero, consistently outperforming Filter Random, which ends with nearly all hard distractors. This asymmetry confirms that the benefit of filtering hard distractors emerges only when their proportion is reduced to near zero; above this threshold, the filtering strategy's benefit is marginal.

**Proportional reduction.** To isolate the pure effect of context length, we shrink context from 131K to 27K tokens while maintaining a fixed hard distractor ratio (20%, 50%, or 80%) throughout. These ratios are chosen to lie beyond the initial first-drop region, where performance has largely entered the saturated regime. At each step, we remove both hard and easy distractors proportionally, for example, we remove 4K tokens of hard distractors and 16K tokens of easy distractors in the 20% setting. In this way, we ensure the composition remains constant as the length decreases.

Figure 9 shows the results. The three curves largely overlap despite varying hard distractor ratios, indicating that performance scales with context length rather than composition. Together with the Filter Hard vs. Filter Random results, this suggests that filtering benefits observed in practice may be primarily a byproduct of context shortening.

This section shows that changing the hard proportion from moderate to high levels has limited marginal effect. In this regime, shortening the context can dominate the observed recovery. The divergence between **Filter Hard** and **Filter Random** at the shortest context lengths should therefore be viewed as an *idealized* boundary case: a clear strategy-specific gain appears only when the hard proportion is pushed close to zero, which is difficult to achieve in realistic retrieval pipelines and is not the regime targeted by most filtering methods. This distinction reconciles the two findings: hard distractor composition has its largest marginal effect near the initial contamination boundary, whereas context length dominates after the context has already been substantially contaminated.

*Table 1.* Experimental design for incremental filtering. Both experiments start at 128K tokens and progressively reduce context length. The symmetric design allows us to separate the effects of context length reduction from distractor composition.

| Context Length | Filter Hard | | Filter Random | |
|---|---|---|---|---|
| | Hard | Random | Hard | Random |
| 131K (start) | 80% | 20% | 20% | 80% |
| 110K | 76% | 24% | 24% | 76% |
| 89K | 71% | 29% | 29% | 71% |
| 69K | 62% | 38% | 38% | 62% |
| 47K | 44% | 56% | 56% | 44% |
| 27K (end) | 3% | 97% | 97% | 3% |

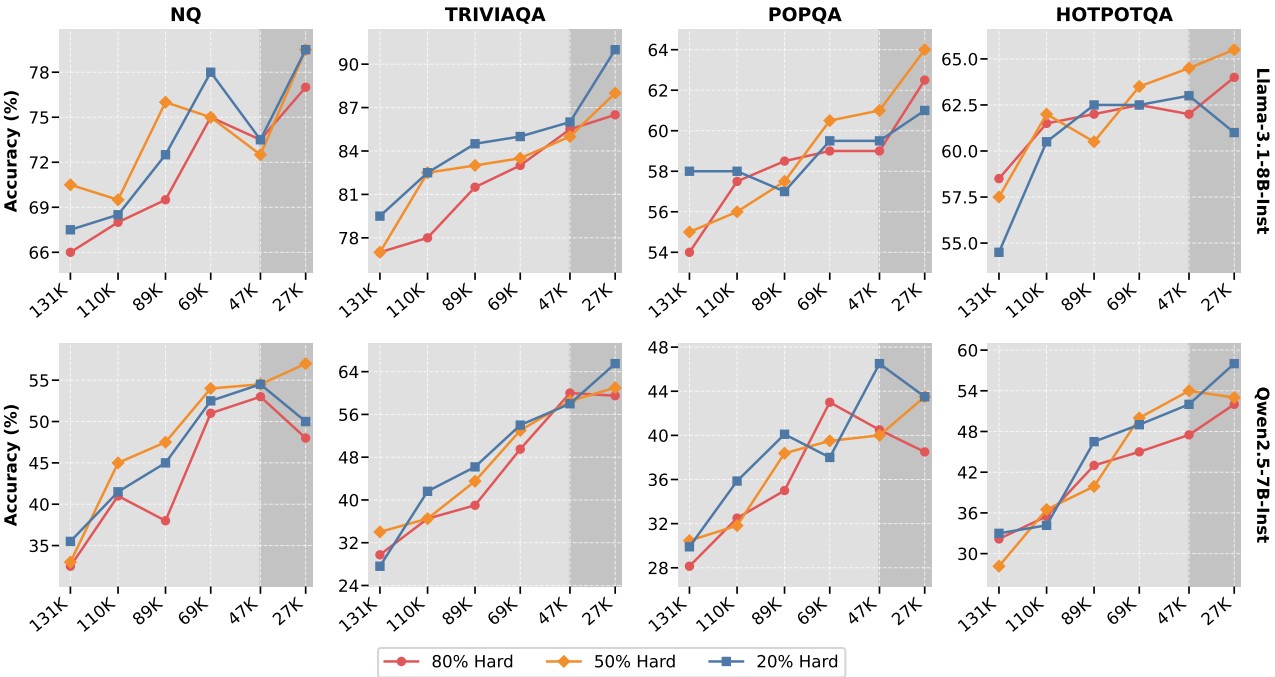

*Figure 9.* **Proportional Reduction.** Context is reduced from 131K to 27K while maintaining a fixed hard distractor ratio (20%, 50%, or 80% hard) by removing documents proportionally from each distractor category. Across both models and all datasets, the three curves follow similar trajectories: reducing the context length consistently improves performance, while varying the fixed hard ratio within this moderate-to-high range has only a limited marginal effect. This suggests that, once the context already contains a non-negligible fraction of hard distractors, the observed recovery is driven primarily by context length reduction rather than by the exact hard-distractor ratio.

## 7. Limitations and Implications

**Limitations.** We use multi-document QA as the experimental setting throughout this paper due to its controllability and ease of evaluation. However, we acknowledge that generalizing our findings to other long-context scenarios (e.g., summarization, code understanding, or multi-turn dialogue) remains an important direction for future work. While we provide both empirical characterization and mechanistic understanding of THE FIRST DROP OF INK effect, we have not yet identified an effective mitigation strategy.

**Implications.** Our work identifies the THE FIRST DROP OF INK effect, implying that removing 90% of hard distractors may recover only a fraction of the lost performance, while the remaining 10% continues to dominate attention. For practitioners, this suggests prioritizing retrieval precision over recall to prevent hard distractors from entering the context in the first place. Additionally, our findings imply that disentangling the effects of context length and distractor composition matters, which prior work often conflates when evaluating long-context models.

## 8. Conclusion

In this work, we identify the THE FIRST DROP OF INK effect: in long-context settings, a small fraction of hard distractors causes disproportionately severe performance degradation, while subsequent additions have diminishing impact. We provide a theoretical explanation grounded in attention mechanics and validate this theory by measuring logit margins on retrieval heads. Our findings challenge the assumption that filtering yields proportional gains and suggest that retrieval precision is far more critical than incremental filtering in long context settings.

## Impact Statement

We do not foresee any direct negative societal consequences of this work. However, we note that improved understanding of attention mechanisms could potentially be misused to craft adversarial inputs; we encourage the community to develop robust defenses alongside mechanistic insights.

## Acknowledgements

We sincerely thank Daniel Khashabi, Taiming Lu, and the Texas A&M NLP community for their helpful comments and feedback.

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

# A. Detailed Experiment Results

As mentioned in §3.2, below are results for all the models across different settings.

*Table 2.* Accuracy (%) of `Llama-3.2-1B-Instruct` across different hard distractor proportions and context lengths. Hard % indicates the proportion of hard distractors, with the remaining being easy distractors (Easy) or random Wikipedia passages (Random).

| Hard % | Easy | | | | | | Random | | | | | |
|---|---|---|---|---|---|---|---|---|---|---|---|---|
| | 4K | 8K | 16K | 32K | 64K | 128K | 4K | 8K | 16K | 32K | 64K | 128K |
| *Natural Questions* | | | | | | | | | | | | |
| 0 | 76.0 | 68.5 | 61.0 | 66.0 | 53.0 | 55.5 | 75.0 | 50.5 | 30.5 | 24.0 | 38.0 | 30.5 |
| 1 | 75.0 | 58.5 | 43.5 | 51.0 | 43.5 | 35.0 | 69.5 | 37.0 | 32.5 | 28.0 | 33.5 | 27.0 |
| 2 | 75.0 | 50.5 | 43.0 | 46.5 | 37.0 | 31.0 | 70.0 | 47.0 | 32.5 | 32.0 | 29.5 | 28.5 |
| 3 | 75.0 | 50.5 | 43.0 | 40.0 | 36.5 | 29.5 | 69.0 | 46.5 | 34.0 | 30.0 | 31.0 | 23.5 |
| 5 | 71.0 | 48.0 | 42.5 | 40.5 | 34.5 | 30.0 | 70.5 | 40.0 | 30.5 | 22.5 | 28.5 | 29.0 |
| 10 | 70.0 | 38.0 | 37.0 | 30.5 | 37.0 | 29.0 | 68.5 | 39.0 | 30.5 | 23.5 | 32.5 | 26.5 |
| 20 | 65.5 | 38.5 | 35.0 | 31.0 | 35.5 | 29.0 | 67.0 | 34.0 | 31.0 | 22.0 | 26.0 | 25.5 |
| 40 | 61.5 | 36.0 | 30.0 | 25.0 | 33.0 | 29.5 | 65.0 | 30.0 | 24.5 | 21.0 | 27.0 | 24.5 |
| 60 | 62.0 | 34.0 | 24.5 | 27.0 | 28.0 | 25.0 | 54.0 | 31.5 | 28.0 | 23.5 | 23.0 | 22.5 |
| 80 | 60.0 | 26.0 | 22.0 | 21.0 | 26.0 | 24.0 | 58.5 | 30.0 | 21.5 | 24.0 | 22.0 | 24.0 |
| 90 | 52.5 | 29.0 | 22.5 | 19.0 | 26.0 | 22.0 | 57.0 | 27.0 | 20.5 | 21.0 | 28.0 | 25.5 |
| 100 | 57.5 | 31.0 | 26.5 | 23.5 | 24.0 | 23.0 | 57.5 | 31.0 | 27.0 | 24.0 | 24.5 | 23.0 |
| *TriviaQA* | | | | | | | | | | | | |
| 0 | 84.5 | 78.0 | 72.5 | 78.5 | 72.0 | 68.5 | 74.5 | 58.5 | 56.0 | 42.0 | 54.5 | 41.5 |
| 1 | 77.5 | 64.5 | 56.0 | 55.5 | 57.0 | 44.5 | 74.0 | 58.0 | 46.5 | 44.0 | 52.0 | 43.5 |
| 2 | 77.5 | 56.5 | 51.5 | 51.0 | 51.0 | 43.5 | 73.0 | 56.5 | 47.5 | 41.5 | 49.5 | 39.0 |
| 3 | 77.5 | 56.5 | 50.5 | 48.0 | 54.0 | 46.5 | 76.0 | 57.5 | 50.0 | 43.5 | 49.5 | 42.5 |
| 5 | 74.0 | 52.0 | 45.5 | 49.5 | 53.5 | 45.0 | 75.5 | 57.5 | 47.0 | 37.5 | 44.5 | 36.5 |
| 10 | 71.5 | 43.0 | 46.0 | 43.0 | 46.5 | 35.0 | 70.5 | 56.0 | 41.5 | 34.0 | 42.0 | 31.0 |
| 20 | 66.5 | 50.5 | 43.0 | 41.0 | 43.0 | 35.5 | 69.5 | 49.5 | 39.0 | 33.0 | 35.0 | 26.5 |
| 40 | 66.5 | 41.5 | 40.0 | 35.5 | 40.5 | 31.0 | 70.0 | 50.0 | 35.0 | 31.5 | 29.5 | 28.0 |
| 60 | 62.0 | 37.5 | 34.0 | 31.0 | 36.0 | 26.0 | 63.5 | 41.0 | 36.5 | 32.0 | 29.5 | 26.0 |
| 80 | 58.5 | 34.0 | 32.5 | 29.0 | 27.0 | 21.5 | 63.0 | 42.5 | 29.5 | 31.5 | 29.0 | 20.5 |
| 90 | 61.0 | 40.5 | 32.5 | 30.0 | 26.5 | 23.5 | 60.5 | 43.0 | 32.5 | 27.0 | 27.5 | 21.5 |
| 100 | 63.5 | 40.5 | 33.0 | 31.0 | 25.0 | 25.0 | 63.5 | 40.5 | 33.0 | 31.0 | 25.5 | 25.0 |
| *PopQA* | | | | | | | | | | | | |
| 0 | 88.5 | 81.0 | 74.0 | 72.0 | 64.0 | 61.5 | 84.0 | 72.5 | 37.5 | 30.5 | 41.0 | 29.5 |
| 1 | 85.0 | 76.0 | 63.0 | 61.5 | 53.5 | 48.0 | 84.0 | 59.5 | 46.0 | 31.0 | 46.0 | 28.5 |
| 2 | 85.0 | 68.0 | 60.0 | 57.0 | 52.5 | 46.5 | 86.5 | 61.5 | 44.0 | 30.0 | 42.5 | 30.0 |
| 3 | 85.0 | 68.0 | 60.5 | 59.0 | 47.0 | 43.0 | 86.5 | 56.5 | 41.5 | 33.5 | 39.0 | 27.5 |
| 5 | 83.5 | 62.5 | 55.0 | 55.0 | 45.5 | 38.0 | 77.5 | 52.0 | 42.5 | 30.0 | 39.0 | 34.5 |
| 10 | 81.0 | 54.0 | 55.0 | 50.0 | 46.0 | 43.5 | 78.5 | 61.0 | 45.0 | 28.5 | 37.5 | 34.5 |
| 20 | 77.5 | 46.5 | 47.5 | 41.0 | 48.0 | 39.0 | 77.5 | 60.0 | 39.0 | 32.0 | 41.5 | 35.5 |
| 40 | 66.0 | 50.5 | 39.5 | 35.0 | 39.5 | 32.0 | 73.5 | 42.0 | 39.5 | 32.5 | 38.0 | 33.0 |
| 60 | 64.5 | 41.0 | 34.0 | 32.0 | 37.0 | 33.5 | 71.0 | 42.5 | 35.5 | 36.0 | 34.5 | 33.5 |
| 80 | 62.5 | 37.0 | 32.5 | 27.5 | 33.5 | 30.5 | 66.5 | 37.0 | 32.5 | 31.0 | 29.0 | 31.0 |
| 90 | 63.0 | 42.0 | 35.0 | 28.5 | 28.5 | 32.5 | 62.0 | 48.0 | 35.0 | 33.5 | 36.5 | 26.5 |
| 100 | 63.5 | 42.0 | 43.0 | 27.5 | 29.5 | 31.5 | 64.0 | 42.0 | 43.0 | 27.5 | 29.5 | 31.5 |
| *HotpotQA* | | | | | | | | | | | | |
| 0 | 64.5 | 56.5 | 53.0 | 58.5 | 54.5 | 51.0 | 53.5 | 48.5 | 37.0 | 29.0 | 35.0 | 26.0 |
| 1 | 59.0 | 48.5 | 45.0 | 44.0 | 44.0 | 32.0 | 52.0 | 45.0 | 37.5 | 31.5 | 37.5 | 23.0 |
| 2 | 59.5 | 48.0 | 45.5 | 46.5 | 46.0 | 28.5 | 52.0 | 46.0 | 36.0 | 26.5 | 35.0 | 23.0 |
| 3 | 59.0 | 48.0 | 42.0 | 45.5 | 41.0 | 31.5 | 52.0 | 46.5 | 32.5 | 31.5 | 36.5 | 22.0 |
| 5 | 54.5 | 43.5 | 34.5 | 41.5 | 41.5 | 26.0 | 49.5 | 43.0 | 36.0 | 28.5 | 30.5 | 20.5 |
| 10 | 54.5 | 43.5 | 37.5 | 33.0 | 33.5 | 26.0 | 51.5 | 43.0 | 37.5 | 25.5 | 33.5 | 19.0 |
| 20 | 51.0 | 45.0 | 34.5 | 32.0 | 31.5 | 20.0 | 48.5 | 45.0 | 28.0 | 23.5 | 28.0 | 21.5 |
| 40 | 49.5 | 39.0 | 29.5 | 25.0 | 30.5 | 15.5 | 46.0 | 38.5 | 33.0 | 22.5 | 25.0 | 19.0 |
| 60 | 42.0 | 37.0 | 29.5 | 26.0 | 22.0 | 22.0 | 46.0 | 39.0 | 31.0 | 20.0 | 23.5 | 17.0 |
| 80 | 44.0 | 35.5 | 27.0 | 23.0 | 23.0 | 19.0 | 43.5 | 35.5 | 26.0 | 23.5 | 24.0 | 17.0 |
| 90 | 43.5 | 35.5 | 24.5 | 23.5 | 25.0 | 19.0 | 42.5 | 35.5 | 29.0 | 22.0 | 24.5 | 21.0 |
| 100 | 43.0 | 33.5 | 27.5 | 23.0 | 25.5 | 18.0 | 43.0 | 33.5 | 27.5 | 22.5 | 26.0 | 18.0 |

*Table 3.* Accuracy (%) of `Llama-3.1-8B-Instruct` across different hard distractor proportions and context lengths. Hard % indicates the proportion of hard distractors, with the remaining being easy distractors (Easy) or random Wikipedia passages (Random).

| Hard % | Easy | | | | | | Random | | | | | |
|---|---|---|---|---|---|---|---|---|---|---|---|---|
| | 4K | 8K | 16K | 32K | 64K | 128K | 4K | 8K | 16K | 32K | 64K | 128K |
| *Natural Questions* | | | | | | | | | | | | |
| 0 | 89.0 | 91.5 | 90.0 | 88.0 | 87.5 | 87.0 | 88.5 | 90.0 | 90.0 | 90.0 | 86.0 | 82.5 |
| 1 | 89.0 | 89.0 | 87.5 | 86.5 | 87.5 | 85.5 | 89.5 | 91.0 | 88.5 | 83.5 | 86.0 | 76.5 |
| 2 | 89.0 | 89.5 | 89.5 | 88.0 | 86.5 | 82.0 | 88.5 | 90.5 | 90.5 | 83.5 | 81.5 | 76.0 |
| 3 | 89.0 | 89.5 | 88.5 | 83.5 | 83.5 | 78.0 | 88.5 | 90.0 | 86.0 | 83.5 | 81.0 | 75.5 |
| 5 | 90.0 | 87.5 | 88.5 | 85.0 | 82.5 | 76.0 | 90.0 | 86.0 | 89.0 | 77.5 | 79.0 | 71.0 |
| 10 | 89.5 | 86.0 | 88.0 | 80.0 | 77.5 | 72.5 | 89.0 | 88.5 | 86.5 | 79.5 | 78.5 | 66.0 |
| 20 | 91.0 | 88.0 | 85.0 | 79.5 | 76.5 | 70.5 | 87.0 | 85.5 | 83.5 | 81.0 | 77.5 | 64.0 |
| 40 | 85.0 | 86.5 | 79.5 | 75.0 | 73.5 | 66.0 | 87.0 | 83.0 | 82.5 | 77.0 | 72.5 | 62.5 |
| 60 | 85.5 | 82.0 | 79.5 | 75.0 | 74.0 | 63.5 | 84.0 | 84.0 | 79.0 | 72.5 | 71.5 | 63.0 |
| 80 | 84.0 | 80.5 | 78.0 | 71.5 | 73.0 | 62.5 | 83.0 | 83.0 | 79.0 | 71.5 | 70.5 | 65.5 |
| 90 | 85.0 | 79.0 | 75.0 | 73.0 | 72.5 | 64.5 | 84.0 | 83.0 | 75.5 | 77.5 | 70.0 | 62.0 |
| 100 | 82.5 | 83.0 | 75.5 | 73.5 | 71.0 | 62.0 | 85.0 | 84.0 | 77.0 | 74.0 | 71.0 | 62.5 |
| *TriviaQA* | | | | | | | | | | | | |
| 0 | 97.0 | 96.5 | 97.0 | 96.0 | 96.5 | 96.0 | 97.5 | 95.5 | 95.0 | 94.0 | 93.5 | 93.5 |
| 1 | 95.0 | 94.5 | 95.0 | 93.0 | 93.0 | 92.5 | 97.0 | 95.0 | 95.5 | 94.5 | 92.5 | 87.0 |
| 2 | 95.0 | 93.5 | 94.5 | 94.0 | 93.5 | 89.0 | 95.5 | 95.0 | 93.5 | 93.0 | 92.0 | 87.5 |
| 3 | 95.0 | 94.0 | 95.0 | 94.5 | 92.0 | 90.5 | 95.0 | 95.5 | 94.0 | 93.0 | 92.0 | 86.0 |
| 5 | 95.0 | 94.0 | 94.0 | 90.0 | 90.0 | 89.5 | 97.0 | 94.0 | 93.0 | 90.5 | 87.5 | 86.0 |
| 10 | 94.0 | 95.0 | 93.0 | 90.5 | 90.5 | 83.5 | 95.5 | 93.0 | 91.5 | 91.0 | 88.0 | 83.5 |
| 20 | 94.0 | 93.5 | 93.0 | 89.0 | 89.0 | 82.5 | 95.5 | 93.5 | 92.5 | 88.5 | 88.0 | 80.0 |
| 40 | 93.5 | 89.5 | 90.5 | 88.0 | 86.5 | 82.0 | 93.5 | 94.0 | 90.0 | 87.5 | 85.0 | 80.0 |
| 60 | 92.5 | 91.0 | 91.0 | 88.0 | 85.5 | 80.0 | 91.0 | 91.5 | 89.5 | 87.5 | 81.5 | 76.0 |
| 80 | 93.0 | 91.5 | 90.0 | 85.5 | 81.5 | 78.0 | 93.0 | 90.5 | 88.0 | 85.0 | 84.0 | 76.0 |
| 90 | 93.0 | 89.5 | 91.0 | 84.5 | 84.0 | 78.0 | 92.0 | 90.0 | 88.5 | 83.5 | 83.5 | 76.0 |
| 100 | 90.0 | 90.0 | 88.0 | 84.5 | 83.5 | 77.5 | 91.0 | 91.0 | 88.0 | 84.0 | 83.5 | 77.0 |
| *PopQA* | | | | | | | | | | | | |
| 0 | 96.0 | 96.0 | 94.5 | 94.0 | 94.5 | 92.0 | 96.0 | 96.5 | 98.0 | 96.0 | 95.0 | 92.5 |
| 1 | 97.0 | 97.0 | 94.5 | 94.5 | 92.5 | 92.5 | 97.0 | 97.5 | 96.0 | 95.0 | 95.5 | 90.0 |
| 2 | 96.5 | 95.5 | 94.5 | 93.0 | 92.0 | 87.5 | 97.5 | 98.0 | 96.5 | 93.0 | 95.5 | 90.0 |
| 3 | 97.5 | 95.5 | 94.5 | 94.0 | 92.0 | 86.0 | 96.0 | 97.5 | 97.0 | 95.5 | 90.0 | 88.5 |
| 5 | 93.5 | 95.5 | 93.0 | 94.0 | 88.0 | 84.0 | 95.0 | 96.5 | 96.5 | 94.0 | 90.0 | 83.5 |
| 10 | 96.0 | 94.5 | 93.0 | 92.0 | 84.0 | 78.5 | 95.5 | 96.0 | 96.0 | 90.5 | 86.0 | 78.5 |
| 20 | 96.0 | 94.5 | 91.5 | 87.0 | 80.5 | 77.5 | 95.5 | 94.0 | 95.0 | 86.5 | 85.0 | 76.0 |
| 40 | 93.0 | 92.5 | 91.0 | 84.5 | 83.0 | 75.5 | 95.0 | 93.5 | 91.5 | 85.5 | 76.0 | 73.0 |
| 60 | 94.0 | 93.0 | 91.0 | 84.0 | 76.0 | 71.0 | 95.0 | 93.0 | 91.5 | 84.5 | 74.0 | 72.5 |
| 80 | 92.5 | 91.0 | 89.5 | 85.5 | 75.5 | 70.5 | 92.0 | 90.5 | 88.5 | 84.0 | 72.0 | 70.5 |
| 90 | 90.0 | 91.5 | 87.5 | 81.5 | 76.5 | 71.5 | 92.0 | 91.5 | 90.0 | 82.0 | 76.0 | 68.0 |
| 100 | 89.0 | 89.0 | 92.0 | 89.0 | 78.0 | 75.0 | 89.0 | 89.0 | 92.0 | 87.0 | 78.0 | 75.0 |
| *HotpotQA* | | | | | | | | | | | | |
| 0 | 86.5 | 85.5 | 83.5 | 80.0 | 81.0 | 70.0 | 80.0 | 71.0 | 67.0 | 60.5 | 61.0 | 49.0 |
| 1 | 85.0 | 82.5 | 76.5 | 71.5 | 71.5 | 60.0 | 74.5 | 67.5 | 67.5 | 61.0 | 61.5 | 51.0 |
| 2 | 85.0 | 76.0 | 72.0 | 69.5 | 71.0 | 57.5 | 77.5 | 71.5 | 65.5 | 60.5 | 62.5 | 49.0 |
| 3 | 85.0 | 76.5 | 72.0 | 68.0 | 66.5 | 57.5 | 73.5 | 71.5 | 65.5 | 61.0 | 61.0 | 44.5 |
| 5 | 81.5 | 74.5 | 69.0 | 65.5 | 66.5 | 51.0 | 74.0 | 68.0 | 64.5 | 56.0 | 62.0 | 45.0 |
| 10 | 77.5 | 74.5 | 68.5 | 60.5 | 63.5 | 47.5 | 75.5 | 64.5 | 63.0 | 57.0 | 59.0 | 43.5 |
| 20 | 75.0 | 70.0 | 64.5 | 58.0 | 60.5 | 47.5 | 75.0 | 65.0 | 60.5 | 55.0 | 58.0 | 38.0 |
| 40 | 70.5 | 66.0 | 61.0 | 61.0 | 62.0 | 46.5 | 74.5 | 65.0 | 58.0 | 54.0 | 60.0 | 41.5 |
| 60 | 69.0 | 65.5 | 61.5 | 56.0 | 60.0 | 44.0 | 70.5 | 64.5 | 59.0 | 57.0 | 54.5 | 45.0 |
| 80 | 70.5 | 65.5 | 58.0 | 58.0 | 60.5 | 45.5 | 71.5 | 63.5 | 54.5 | 54.0 | 55.5 | 41.5 |
| 90 | 73.0 | 65.5 | 61.0 | 57.0 | 59.0 | 45.5 | 68.5 | 63.5 | 58.0 | 55.5 | 56.0 | 44.0 |
| 100 | 67.0 | 64.0 | 57.0 | 58.0 | 62.0 | 48.0 | 67.0 | 64.0 | 57.0 | 58.0 | 63.0 | 48.0 |

*Table 4.* Accuracy (%) of `Qwen2.5-7B-Instruct` across different hard distractor proportions and context lengths. Hard % indicates the proportion of hard distractors, with the remaining being easy distractors (Easy) or random Wikipedia passages (Random).

| Hard % | Easy | | | | | | Random | | | | | |
|---|---|---|---|---|---|---|---|---|---|---|---|---|
| | 4K | 8K | 16K | 32K | 64K | 128K | 4K | 8K | 16K | 32K | 64K | 128K |
| *Natural Questions* | | | | | | | | | | | | |
| 0 | 84.5 | 83.5 | 75.5 | 75.0 | 75.5 | 68.0 | 78.5 | 71.5 | 70.5 | 74.5 | 70.0 | 44.5 |
| 1 | 82.0 | 78.0 | 75.5 | 73.0 | 71.0 | 69.0 | 72.0 | 71.5 | 69.5 | 64.0 | 60.5 | 35.0 |
| 2 | 81.5 | 76.0 | 74.5 | 70.5 | 71.5 | 52.5 | 80.5 | 73.5 | 70.5 | 65.0 | 58.0 | 36.5 |
| 3 | 83.5 | 77.0 | 73.0 | 72.5 | 68.0 | 51.5 | 74.0 | 69.0 | 68.5 | 61.5 | 62.0 | 32.0 |
| 5 | 80.0 | 79.5 | 72.0 | 72.0 | 67.5 | 42.5 | 70.5 | 72.0 | 72.0 | 65.5 | 63.0 | 36.5 |
| 10 | 82.0 | 75.0 | 69.0 | 65.5 | 65.0 | 45.5 | 72.0 | 68.0 | 72.5 | 64.5 | 58.0 | 32.5 |
| 20 | 77.0 | 71.0 | 69.5 | 58.5 | 62.0 | 34.0 | 74.5 | 72.0 | 68.0 | 54.5 | 55.5 | 35.5 |
| 40 | 75.0 | 69.0 | 67.5 | 53.0 | 54.0 | 41.0 | 75.0 | 69.0 | 66.0 | 54.5 | 51.5 | 33.0 |
| 60 | 76.0 | 67.0 | 61.5 | 48.5 | 57.0 | 40.0 | 76.5 | 67.0 | 66.5 | 53.5 | 50.0 | 33.5 |
| 80 | 73.0 | 66.5 | 62.0 | 57.0 | 51.0 | 30.5 | 73.0 | 63.5 | 62.5 | 47.0 | 56.0 | 32.5 |
| 90 | 73.5 | 67.5 | 65.0 | 53.5 | 49.0 | 29.5 | 71.0 | 66.5 | 60.5 | 45.5 | 46.0 | 33.0 |
| 100 | 74.0 | 63.0 | 62.0 | 55.5 | 54.5 | 32.5 | 73.5 | 64.5 | 61.0 | 55.0 | 54.0 | 32.0 |
| *TriviaQA* | | | | | | | | | | | | |
| 0 | 89.0 | 87.5 | 89.5 | 83.0 | 82.0 | 81.0 | 88.5 | 84.0 | 78.5 | 74.5 | 67.0 | 42.5 |
| 1 | 86.5 | 84.0 | 79.0 | 75.0 | 73.0 | 63.0 | 86.0 | 79.5 | 78.0 | 74.0 | 65.0 | 39.0 |
| 2 | 86.5 | 84.5 | 82.5 | 74.5 | 69.5 | 56.5 | 85.5 | 81.5 | 80.0 | 72.0 | 65.0 | 31.5 |
| 3 | 86.5 | 83.5 | 79.5 | 71.5 | 66.5 | 53.0 | 87.0 | 80.0 | 76.5 | 73.0 | 61.0 | 34.5 |
| 5 | 85.5 | 81.5 | 76.0 | 67.5 | 64.0 | 44.5 | 87.0 | 80.5 | 78.5 | 68.0 | 66.5 | 35.0 |
| 10 | 81.0 | 79.0 | 76.5 | 64.0 | 59.0 | 41.0 | 86.0 | 74.5 | 74.5 | 70.5 | 66.5 | 34.0 |
| 20 | 83.0 | 76.5 | 76.5 | 65.5 | 58.5 | 44.0 | 81.5 | 74.0 | 75.0 | 59.5 | 58.0 | 27.5 |
| 40 | 76.0 | 74.0 | 69.5 | 58.5 | 57.5 | 34.5 | 80.5 | 74.0 | 74.0 | 64.5 | 55.0 | 29.5 |
| 60 | 77.0 | 70.5 | 72.0 | 56.0 | 53.5 | 34.0 | 77.0 | 71.0 | 73.0 | 55.5 | 52.0 | 30.5 |
| 80 | 75.0 | 74.0 | 70.5 | 63.5 | 52.0 | 33.5 | 72.5 | 70.5 | 69.0 | 62.5 | 48.5 | 30.0 |
| 90 | 74.5 | 73.5 | 68.5 | 60.5 | 58.0 | 32.5 | 74.5 | 72.5 | 67.5 | 60.0 | 54.5 | 28.0 |
| 100 | 74.0 | 70.5 | 70.0 | 59.5 | 57.0 | 32.0 | 74.5 | 70.0 | 70.0 | 59.0 | 56.5 | 32.5 |
| *PopQA* | | | | | | | | | | | | |
| 0 | 93.0 | 93.0 | 88.5 | 88.5 | 89.5 | 86.0 | 91.0 | 89.5 | 82.0 | 82.0 | 74.5 | 91.0 |
| 1 | 90.0 | 92.5 | 83.0 | 87.0 | 83.0 | 68.0 | 91.5 | 87.5 | 86.0 | 77.0 | 71.5 | 41.0 |
| 2 | 89.0 | 88.0 | 85.0 | 83.5 | 75.0 | 60.0 | 90.0 | 88.0 | 85.5 | 77.5 | 71.0 | 39.5 |
| 3 | 89.5 | 87.5 | 85.0 | 81.0 | 70.5 | 61.5 | 91.5 | 87.5 | 83.5 | 77.0 | 67.0 | 34.0 |
| 5 | 88.0 | 88.5 | 87.0 | 77.5 | 72.0 | 48.0 | 89.5 | 86.0 | 83.5 | 76.5 | 63.5 | 36.0 |
| 10 | 89.0 | 81.5 | 84.5 | 76.0 | 70.5 | 47.0 | 89.0 | 81.5 | 80.0 | 72.5 | 62.0 | 42.0 |
| 20 | 89.0 | 83.0 | 80.0 | 72.0 | 63.5 | 45.0 | 88.5 | 79.5 | 79.5 | 73.0 | 64.0 | 32.0 |
| 40 | 84.5 | 83.0 | 80.0 | 71.0 | 62.0 | 37.0 | 87.5 | 81.5 | 75.5 | 65.0 | 55.5 | 32.5 |
| 60 | 82.0 | 78.0 | 71.5 | 63.5 | 53.0 | 38.0 | 87.0 | 76.5 | 72.5 | 63.0 | 51.5 | 33.5 |
| 80 | 83.5 | 76.5 | 74.5 | 60.0 | 54.5 | 34.5 | 84.5 | 71.0 | 76.0 | 60.0 | 49.5 | 27.0 |
| 90 | 80.0 | 77.5 | 75.5 | 61.0 | 50.0 | 27.5 | 85.5 | 76.0 | 77.0 | 60.5 | 46.5 | 29.5 |
| 100 | 84.0 | 70.0 | 73.0 | 58.5 | 50.5 | 28.0 | 84.0 | 70.5 | 72.5 | 57.5 | 53.0 | 29.0 |
| *HotpotQA* | | | | | | | | | | | | |
| 0 | 75.5 | 76.0 | 77.0 | 73.5 | 70.0 | 68.5 | 71.0 | 67.0 | 63.5 | 62.5 | 50.5 | 24.5 |
| 1 | 73.5 | 73.0 | 72.5 | 67.5 | 56.0 | 52.0 | 68.5 | 62.0 | 65.0 | 63.0 | 53.5 | 26.5 |
| 2 | 75.0 | 75.5 | 65.5 | 60.5 | 56.5 | 44.0 | 68.5 | 61.5 | 63.0 | 56.5 | 51.5 | 29.5 |
| 3 | 74.5 | 73.5 | 67.5 | 61.5 | 55.5 | 39.0 | 67.5 | 65.5 | 56.0 | 60.5 | 53.0 | 25.0 |
| 5 | 74.0 | 69.0 | 68.0 | 59.0 | 50.0 | 39.5 | 67.0 | 63.0 | 58.0 | 56.5 | 45.5 | 26.5 |
| 10 | 73.0 | 64.0 | 65.0 | 55.5 | 49.0 | 35.0 | 62.0 | 62.5 | 60.0 | 53.5 | 45.5 | 21.0 |
| 20 | 63.5 | 63.5 | 61.0 | 54.5 | 44.5 | 27.5 | 67.0 | 57.0 | 48.0 | 45.0 | 44.0 | 22.0 |
| 40 | 68.5 | 55.0 | 52.0 | 48.5 | 43.0 | 24.0 | 64.5 | 54.5 | 53.0 | 46.5 | 37.5 | 22.0 |
| 60 | 65.0 | 58.0 | 53.5 | 44.0 | 38.0 | 24.5 | 61.5 | 55.0 | 48.5 | 44.5 | 41.0 | 23.5 |
| 80 | 69.5 | 53.5 | 48.5 | 44.5 | 42.5 | 24.5 | 65.5 | 49.5 | 47.0 | 42.5 | 38.5 | 23.5 |
| 90 | 64.0 | 55.5 | 49.5 | 41.0 | 45.0 | 22.0 | 64.0 | 53.0 | 44.5 | 38.5 | 41.5 | 21.0 |
| 100 | 66.5 | 56.5 | 48.0 | 40.0 | 39.5 | 18.5 | 67.0 | 58.0 | 48.0 | 41.0 | 38.5 | 19.0 |

*Table 5.* Accuracy (%) of `Qwen3-Next-80B-Instruct` across different hard distractor proportions and context lengths. Hard % indicates the proportion of hard distractors, with the remaining being easy distractors (Easy) or random Wikipedia passages (Random).

| | Easy | | | | | | Random | | | | | |
|---|---|---|---|---|---|---|---|---|---|---|---|---|
| Hard % | 4K | 8K | 16K | 32K | 64K | 128K | 4K | 8K | 16K | 32K | 64K | 128K |
| *Natural Questions* | | | | | | | | | | | | |
| 0 | 93.0 | 92.0 | 93.5 | 92.0 | 91.0 | 91.5 | 94.0 | 90.5 | 90.0 | 89.5 | 90.5 | 91.5 |
| 1 | 92.0 | 92.0 | 91.0 | 90.5 | 87.0 | 88.0 | 91.5 | 91.5 | 90.0 | 89.0 | 90.5 | 87.0 |
| 2 | 91.5 | 92.0 | 90.5 | 87.5 | 87.0 | 89.5 | 90.0 | 90.0 | 90.0 | 90.5 | 90.5 | 85.0 |
| 3 | 92.0 | 92.0 | 90.0 | 88.0 | 88.5 | 83.0 | 92.5 | 88.5 | 90.5 | 88.0 | 89.5 | 85.5 |
| 5 | 92.0 | 90.5 | 85.5 | 89.5 | 86.0 | 87.0 | 90.0 | 89.5 | 90.5 | 88.5 | 87.0 | 85.5 |
| 10 | 91.5 | 88.5 | 88.5 | 86.5 | 87.0 | 84.5 | 91.0 | 86.5 | 87.5 | 85.5 | 85.0 | 82.0 |
| 20 | 91.0 | 90.5 | 88.5 | 88.0 | 84.5 | 81.5 | 89.0 | 88.5 | 86.5 | 85.5 | 85.5 | 82.0 |
| 40 | 91.0 | 89.5 | 86.5 | 83.5 | 83.5 | 82.0 | 87.0 | 87.5 | 89.5 | 84.0 | 81.5 | 79.5 |
| 60 | 88.5 | 88.0 | 84.5 | 84.5 | 83.5 | 79.5 | 86.5 | 88.0 | 85.0 | 84.0 | 82.0 | 78.5 |
| 80 | 88.0 | 87.0 | 86.5 | 83.0 | 80.0 | 79.0 | 87.5 | 88.5 | 81.0 | 82.5 | 81.5 | 78.0 |
| 90 | 87.5 | 87.5 | 82.5 | 84.5 | 83.5 | 80.0 | 87.0 | 86.0 | 83.5 | 83.5 | 81.5 | 77.0 |
| 100 | 88.0 | 85.0 | 84.0 | 83.5 | 82.5 | 77.5 | 88.0 | 84.5 | 85.5 | 84.0 | 80.5 | 76.5 |
| *TriviaQA* | | | | | | | | | | | | |
| 0 | 99.5 | 99.5 | 99.5 | 99.5 | 99.5 | 99.5 | 98.0 | 97.5 | 98.0 | 98.5 | 98.5 | 98.0 |
| 1 | 98.0 | 98.5 | 98.0 | 96.5 | 96.0 | 95.5 | 95.5 | 97.5 | 97.0 | 98.0 | 97.0 | 96.5 |
| 2 | 98.5 | 97.0 | 97.5 | 95.5 | 96.5 | 94.0 | 97.0 | 97.0 | 97.0 | 97.0 | 96.5 | 95.0 |
| 3 | 98.5 | 97.5 | 96.5 | 94.0 | 95.0 | 95.0 | 95.5 | 97.0 | 96.0 | 97.5 | 96.0 | 94.0 |
| 5 | 97.5 | 98.0 | 96.5 | 94.0 | 97.0 | 96.0 | 95.0 | 96.5 | 96.0 | 96.5 | 96.0 | 93.5 |
| 10 | 96.5 | 95.5 | 94.0 | 94.5 | 95.0 | 95.5 | 96.5 | 94.5 | 94.5 | 95.0 | 93.5 | 94.0 |
| 20 | 96.5 | 94.5 | 94.5 | 94.0 | 94.5 | 93.0 | 95.0 | 95.0 | 96.0 | 94.5 | 96.5 | 93.0 |
| 40 | 95.0 | 95.5 | 95.5 | 93.0 | 94.0 | 92.0 | 94.0 | 95.5 | 94.5 | 94.0 | 93.0 | 92.5 |
| 60 | 95.5 | 95.5 | 93.0 | 94.5 | 92.0 | 89.5 | 96.5 | 94.0 | 94.5 | 93.0 | 92.5 | 89.0 |
| 80 | 96.0 | 96.0 | 95.0 | 92.0 | 93.5 | 89.0 | 94.5 | 95.0 | 93.5 | 92.5 | 90.0 | 85.5 |
| 90 | 95.5 | 94.0 | 92.0 | 92.5 | 92.5 | 88.0 | 96.0 | 92.0 | 94.5 | 92.0 | 91.0 | 86.5 |
| 100 | 93.5 | 94.5 | 94.0 | 90.5 | 90.0 | 87.0 | 95.0 | 93.0 | 95.0 | 91.5 | 90.5 | 87.0 |
| *PopQA* | | | | | | | | | | | | |
| 0 | 97.5 | 97.5 | 98.0 | 98.0 | 98.0 | 98.0 | 98.5 | 98.5 | 99.0 | 98.5 | 98.5 | 97.5 |
| 1 | 98.0 | 98.0 | 97.0 | 96.5 | 97.5 | 97.0 | 98.5 | 98.5 | 98.0 | 97.0 | 97.5 | 95.0 |
| 2 | 97.5 | 97.5 | 97.0 | 97.5 | 97.0 | 96.0 | 98.0 | 97.5 | 97.0 | 97.5 | 96.5 | 94.0 |
| 3 | 97.5 | 97.5 | 97.0 | 97.5 | 97.0 | 94.5 | 98.0 | 98.5 | 98.5 | 97.0 | 96.5 | 92.0 |
| 5 | 96.5 | 97.5 | 97.5 | 97.5 | 97.5 | 93.0 | 97.0 | 97.5 | 98.5 | 96.0 | 96.0 | 92.0 |
| 10 | 96.5 | 95.5 | 97.0 | 93.5 | 95.0 | 94.0 | 97.0 | 97.5 | 97.0 | 95.5 | 93.5 | 89.5 |
| 20 | 97.5 | 96.5 | 95.0 | 94.5 | 93.0 | 87.5 | 97.5 | 97.0 | 96.5 | 94.5 | 91.0 | 89.0 |
| 40 | 98.5 | 95.5 | 91.0 | 89.5 | 88.0 | 86.0 | 97.0 | 96.0 | 90.5 | 91.0 | 88.5 | 84.5 |
| 60 | 97.0 | 94.5 | 91.0 | 92.0 | 86.5 | 84.5 | 98.0 | 94.0 | 91.5 | 88.0 | 87.0 | 81.5 |
| 80 | 96.0 | 93.5 | 93.5 | 86.0 | 86.0 | 82.5 | 97.0 | 92.0 | 92.5 | 87.5 | 85.0 | 78.5 |
| 90 | 97.0 | 92.5 | 93.0 | 88.5 | 86.0 | 81.5 | 96.0 | 91.5 | 92.5 | 85.5 | 84.5 | 78.5 |
| 100 | 96.0 | 92.5 | 91.0 | 87.0 | 84.0 | 80.0 | 95.0 | 92.0 | 91.5 | 87.5 | 84.0 | 81.5 |
| *HotpotQA* | | | | | | | | | | | | |
| 0 | 88.5 | 90.5 | 88.0 | 89.0 | 90.5 | 90.0 | 85.0 | 84.5 | 82.0 | 81.0 | 82.0 | 69.0 |
| 1 | 85.0 | 87.5 | 86.0 | 85.0 | 88.5 | 84.5 | 85.0 | 84.5 | 79.5 | 81.0 | 80.0 | 69.0 |
| 2 | 85.5 | 88.0 | 81.5 | 83.5 | 84.5 | 82.0 | 83.0 | 85.0 | 80.0 | 80.5 | 77.5 | 69.5 |
| 3 | 85.5 | 88.0 | 82.0 | 84.5 | 84.5 | 78.5 | 85.0 | 85.5 | 79.5 | 82.0 | 81.0 | 68.5 |
| 5 | 88.0 | 84.0 | 79.5 | 81.0 | 82.5 | 75.0 | 85.0 | 84.0 | 78.0 | 79.0 | 79.5 | 67.0 |
| 10 | 86.5 | 85.0 | 78.5 | 80.0 | 80.5 | 76.0 | 85.0 | 84.0 | 77.5 | 76.0 | 76.5 | 67.0 |
| 20 | 83.5 | 83.0 | 75.5 | 75.0 | 79.0 | 68.5 | 81.0 | 82.0 | 75.5 | 74.0 | 74.0 | 64.0 |
| 40 | 83.0 | 80.5 | 73.0 | 74.0 | 77.0 | 70.0 | 83.0 | 81.5 | 75.0 | 71.0 | 72.5 | 62.0 |
| 60 | 82.5 | 77.0 | 72.0 | 74.0 | 74.5 | 67.5 | 82.5 | 79.0 | 73.5 | 70.5 | 67.0 | 64.5 |
| 80 | 81.0 | 79.5 | 73.5 | 73.5 | 76.0 | 67.0 | 82.0 | 81.5 | 73.0 | 71.0 | 69.5 | 67.0 |
| 90 | 80.5 | 80.0 | 73.0 | 74.0 | 71.5 | 68.5 | 79.0 | 80.5 | 72.5 | 72.0 | 70.0 | 64.0 |
| 100 | 81.5 | 79.0 | 71.5 | 73.0 | 71.5 | 69.0 | 81.5 | 79.5 | 72.5 | 74.5 | 72.0 | 69.5 |

# B. Margin Computation ($\Delta_e$ and $\Delta_h$)

As mentioned in §5, we calculate the margin for `Llama-3.1-8b-Instruct` and `Llama-3.2-1b-Instruct`. Below are results for `Llama-3.2-1b-Instruct` and the correlation results for both models.

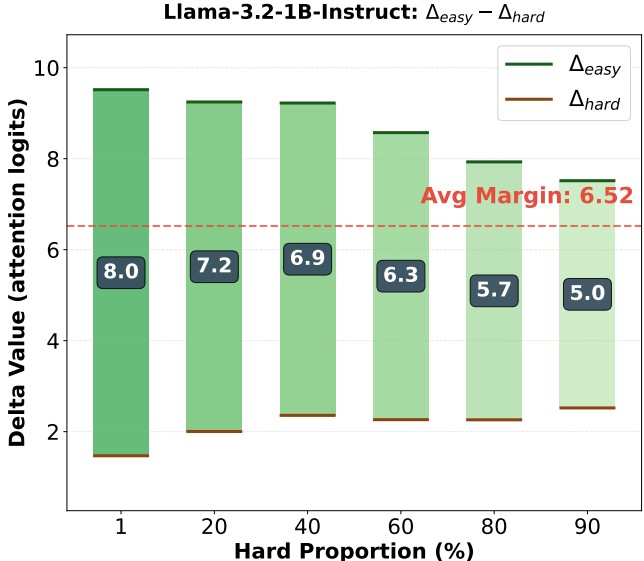

*Figure 10.* Empirical measurement of logit margins $\Delta_e$ and $\Delta_h$ on retrieval heads for Llama-3.2-1B-Instruct. Green bars show $\Delta_e$ (margin to easy distractors) and brown bars show $\Delta_h$ (margin to hard distractors). The gap $\Delta_e - \Delta_h$ remains substantial across all hard proportions, with an average of 6.52, which is more significant than the gap of the 8B model. This validates the theoretical assumption $\Delta_h \ll \Delta_e$.

*Table 6.* Per-file train–test correlations for selected hard proportions on `nq_easy`.

| Hard Proportion | 8B Pearson | 8B Spearman | 1B Pearson | 1B Spearman |
|---|---|---|---|---|
| 1% | 0.9498 | 0.9838 | 0.9599 | 0.9953 |
| 20% | 0.9619 | 0.9854 | 0.9592 | 0.9962 |
| 40% | 0.9753 | 0.9915 | 0.9546 | 0.9970 |
| 60% | 0.9725 | 0.9894 | 0.9522 | 0.9964 |
| 80% | 0.9508 | 0.9832 | 0.9522 | 0.9922 |
| 90% | 0.9491 | 0.9850 | 0.9278 | 0.9868 |

# C. Prompts Demonstration

In this section, we demonstrate the prompts used for: (1)evaluating model's output and (2) verifying the distractors not containing the answers.

---

**Evaluation Prompt**

You are an expert evaluator for question-answering systems. Your task is to determine if a model's answer to a question is correct based on the provided documents and reference answer.

**# Documents**
{context}

**# Question**
{question}

**# Reference Answer**
{reference_answer}

**# Model's Answer**
{model_answer}

**# Evaluation Task**
Based on the information in the provided documents and the reference answer, evaluate whether the model's answer is correct.

The answer is **CORRECT** if:
1. It matches or is semantically equivalent to the reference answer
2. It accurately answers the question using information from the documents
3. It does not contain extra hallucinated or incorrect information

The answer should be considered **CORRECT** even if:
- It uses slightly different wording but conveys the same meaning
- It uses synonyms or alternative names for the same entity
- It is a shorter or longer form of the reference answer (e.g., "Steven Weber" vs "Steven Robert Weber")

Respond with **ONLY** one of these two options:
- CORRECT
- INCORRECT

---

**Distractor Verification Prompt**

You are evaluating whether a set of documents contains the answer to a given question.

**# Question**
{question}

**# Accepted Answers**
{', '.join(answer)}

**# Documents**
{docs_text}

**# Task**
Determine if ANY of the documents above EXPLICITLY contain information that would allow someone to answer the question with one of the accepted answers. Please respond in the following JSON format:

```
{
    "has_answer": true/false,
    "explanation": "Brief explanation...",
    "confidence": "high/medium/low"
}
```

**# Critical Rules**
- You must ONLY use information that is EXPLICITLY stated in the provided documents
- Do NOT use any external knowledge or make inferences based on your own knowledge
- Do NOT assume facts that are not directly written in the documents
- Answer "has_answer": true ONLY if the answer is EXPLICITLY written in the documents
- Consider synonyms and paraphrases (e.g., "USA" and "United States" are equivalent)
- If the documents mention the topic but don't EXPLICITLY contain the specific answer, count it as false

**# Examples**
- INVALID reasoning: "Green Day is known for punk rock" - this uses external knowledge
- VALID reasoning: "The document states 'punk rock band Green Day'" - this quotes the document

