# OpenReview forum: "The First Drop of Ink: Nonlinear Impact of Misleading Information in Long-Context Reasoning"
_ICML.cc/2026/Conference — ICML 2026 regular_

### Official Review · Reviewer_HLu4 · 2026-03-12

**Soundness:** 3
**Presentation:** 3
**Significance:** 3
**Originality:** 3
**Overall Recommendation:** 5
**Confidence:** 4

**Summary:**

This work explores how the proportion of semantically misleading ("hard") distractors affects LLM performance in long-context multi-document QA.
The central finding is the "First Drop of Ink" effect, which means that performance degrades sharply when even a small fraction of hard distractors is introduced (10%), while further increases cause only marginal additional decline.
The authors provide a theoretical explanation grounded in the convexity of softmax attention, empirically validate it by measuring logit margins on retrieval heads, and analyze the implications for filtering strategies.

**Compliance With Llm Reviewing Policy:**

Affirmed.

**Final Justification:**

Thanks for your response. I will keep my score.

**Key Questions For Authors:**

See Weaknesses.

**Limitations:**

yes

**Strengths And Weaknesses:**

### Strengths
1. Clear and well-motivated phenomenon. The nonlinear degradation pattern is empirically robust across four datasets, four models, and six context lengths. The "drop ratio" metric effectively quantifies the front-loaded nature of degradation, and Figure 3 makes the cross-model consistency convincing.
2. Lemma 4.1 and 4.2 provide a clean derivation showing that attention on the gold document is a strictly convex, strictly decreasing function of hard distractor proportion. The simplified form in Remark 4.3 gives intuitive insight into what controls the curve's vertical position versus shape, and the empirical validation via retrieval head logit margins in §5 ties theory to practice effectively.
3. The Filter Hard vs. Filter Easy and Proportional Reduction experiments in §6.2 are methodologically strong. Showing that filtering gains are primarily attributable to context length reduction—not distractor removal—is an important and counterintuitive finding that challenges a common assumption in the RAG literature.

### Weaknesses
1. It would be better to have more analysis about how would different models affected by the "First Drop of Ink" effect. Specifically, Figure 2 and 3 show substantially flatter degradation curves for Qwen3-Next-80B-Instruct, especially on HotpotQA and TriviaQA. This suggests the effect may be model-capacity-dependent, but this difference is not analyzed or theoretically accounted for. Understanding what architectural or training differences attenuate the effect would be valuable.

---

> ### Author Rebuttal · Authors · 2026-03-31
>
> We thank the reviewer for recognizing our empirical contribution ("Clear and well-motivated"), theoretical analysis ("a clean derivation") and experiment design ("methodologically strong"). Below we clarify the rest concern.
>
> ---
> - How would different models affected by "First Drop of Ink" effect.
>
> We agree with the reviewer that understanding cross-model variation is valuable. Qwen3-Next-80B adopts a hybrid attention design of Gated DeltaNet and standard softmax attention. The gating mechanism introduces a learned multiplicative control after attention computation, which can selectively suppress distractor contributions even when softmax has already allocated attention to them. We therefore conjecture that gated attention is a potential direction for future work.
>
> That said, even with this architectural advantage, the First Drop of Ink effect remains clearly present in Qwen3-Next across all datasets. We thank the reviewer for raising this point and will include this discussion in the revision.
>
> [1] Qiu et al., "Gated Attention for Large Language Models: Non-linearity, Sparsity, and Attention-Sink-Free", NeurIPS 2025.

---

> > ### Author Rebuttal · Reviewer_HLu4 · 2026-04-02
> >
> > Thanks for your response. I will keep my score.

---

> > > ### Author Response · Authors · 2026-04-04
> > >
> > > We truly appreciate your effort in reviewing our paper and the suggestion about model architecture impact on the First Drop of Ink effect. Thank you for your time and valuable feedback!

---

### Official Review · Reviewer_g96m · 2026-03-13

**Soundness:** 3
**Presentation:** 3
**Significance:** 3
**Originality:** 3
**Overall Recommendation:** 5
**Confidence:** 4

**Summary:**

This paper investigates the impact of hard negative distractors on RAG performance. The main finding is that the relationship is highly nonlinear: the first 10% of hard distractors causes 58% of total performance degradation. The paper provides a retrieval-head-based theoretical explanation grounded in the attention margin gap between easy and hard distractors, and validates it empirically. A filtering analysis further shows that the benefit of removing distractors comes primarily from reducing context length rather than from eliminating hard negatives. Experiments span 4 datasets, 4 models, and 6 context lengths.

**Compliance With Llm Reviewing Policy:**

Affirmed.

**Final Justification:**

All major concerns addressed with new evidence: bootstrap CIs confirm anomalies are noise, post-softmax attention validates the theory, and extended filtering experiments generalize across models and datasets. I raise my score to 5.

**Key Questions For Authors:**

**Q1.** How do the authors explain the drop ratio values above 1 and below 0 in Figure 3? If these are just noise (e.g., small sample sizes), showing error bars or confidence intervals would resolve my concern in W2. If they reflect real non-monotonicity, the theoretical model in Lemma 4.2 needs to account for it. This is the question most likely to change my assessment of the paper's soundness.

**Q2.** Can the authors report the post-softmax attention mass on the gold document as a function of hard proportion? The current validation in logit space (W3) leaves a gap between what is measured and what the theory actually claims. Showing the post-softmax numbers would directly confirm or challenge the proposed mechanism.

**Q3.** The filtering experiments only use Llama-3.1-8B on nq_random. Do the conclusions hold for other models and datasets? This would affect how strongly I weigh the filtering analysis as a contribution.

**Limitations:**

yes

**Strengths And Weaknesses:**

### **Strengths**:

S1. They reveals two counterintuitive findings: the first 10% of hard distractors causes 58% of total degradation, and filtering
  helps mainly by shortening context rather than removing distractors.

S2. The symmetric Filter Hard vs. Filter Easy setup cleanly separates the effect of context length from distractor composition, providing solid causal evidence.

S3. The key theoretical assumption is directly measured on retrieval heads rather than just stated.

S4. Results hold across 4 datasets, 4 models, and 6 context lengths


### **Weaknesses**:

W1. Two related papers missing or under-discussed.
 1. *Jin et al., "Long-Context LLMs Meet RAG: Overcoming Challenges for Long Inputs in RAG" (ICLR 2025)*  study hard negatives on the same datasets and observe an inverted-U pattern, while this paper shows monotonic decline. The contradiction is not addressed.
 2.  *Du et al., "Context Length Alone Hurts LLM Performance Despite Perfect Retrieval" (EMNLP 2025)* independently show that context  length alone hurts performance, overlapping substantially with Section 6.2.

W2. Unexplained anomalies in the drop ratio.

Figure 3 contains values above 1 and negative values. Values above 1 imply non-monotonicity, directly contradicting Lemma 4.2. The paper does not acknowledge or explain these cases. If non-monotonicity exists in practice, then the theoretical model is incomplete.

W3. Gap between theory and validation.

The theory operates on post-softmax attention, but validation uses pre-softmax logits. The justification (numericalbunderflow at 128K) is practical, not theoretical. More importantly, the ratio $b/a = e^{\Delta_e - \Delta_h} \approx 340$ only matters if hard distractors actually steal meaningful attention from the gold document. Without reporting the absolute logit gap to the gold document or actual post-softmax attention mass, the mechanism is not fully validated.


W4. No concrete mitigation proposed.

Temperature scaling fails. Filtering is diagnostic. The suggestion to "improve upstream retrieval precision" is too vague. This is acceptable for an analysis paper, but even one specific mitigation direction would make the paper stronger.

---

> ### Author Rebuttal · Authors · 2026-03-31
>
> We thank the reviewer for recognizing our contribution (“providing solid causal evidence”, “clearly separates”). Below we clarify the weakness and questions.
>
> ---
>
> - W1 Related work
>
> **TLDR — Neither contradicts our claim nor diminishes our contribution.**
>
> We thank the reviewer for pointing out these two highly relevant works. Jin et al. has already been in the related work, and we will further expand our discussion of it in the revision. We will also include and discuss Du et al.
>
> For Jin et al., we would like to clarify a very important difference in the context scale (**4K vs.128K**). Most of their experiments use fewer than 40 passages from a Wikipedia dump (~100-150 tokens each), placing their total context length at approximately 4-6K tokens. Further, their Mistral experiments (Fig 5) that extend to longer contexts also show monotonic decline, consistent with our findings.
>
> For Du et al., their finding that context length alone degrades performance is consistent with our $\S$6. However, we respectfully argue our finding is non-trivial and goes further.  Du et al. setup extends length with non-relevant fillers (essays, whitespace, or attention masking), which does not address a more practical question: when the context contains a mixture of genuinely misleading hard distractors, does filtering them out actually help? Our work shows the answer is largely no, and filtering gains come primarily from length regardless of which type is removed. This practical insight is not addressable within Du et al.'s framework.
>
> In short, Jin et al. operates at a much shorter scale with minimal overlap, and Du et al. addresses length effects but not distractor composition. Neither contradicts or covers our finding.
>
> ---
>
> - W2 & Q1 The Drop Ratio
>
> **TLDR—We add confidence intervals to confirm the outliers are noise and rerun the affected settings with more samples; all yield normal drop ratios.**
>
> We appreciate the reviewer for this suggestion. We compute 95% bootstrap confidence intervals for all drop ratios in the original Figure 3 (https://anonymous.4open.science/r/89XLms/fig3_ci.pdf).
>
> All anomalous values share a common characteristic of extremely wide CIs compared to normal cells (~3.0 vs. ~0.2), confirming these are noise. We rerun the outlier settings in Llama-3.1-8B-Instruct under NQ and HotpotQA with different seeds, and the results are all well within the expected range and consistent with the nonlinear pattern (0.22, 0.16, and 0.80).
>
> We will (1) update Fig3 with confidence intervals and (2) release all QA pairs and distractor collections to facilitate future research and reproducibility.
>
> ---
>
> - W3 & Q2 Post-softmax attention mass
>
> **TLDR—Post-softmax attention curve directly confirms the convex degradation predicted by our theory.**
>
> Below we present the post-softmax attention mass on the gold document for Llama-3.1-8B-Instruct on nq_random at 128K context (50 samples per data point):
>
> |      Hard %      |      1%     |     20%    |     40%    |     60%    |     80%    |     90%    |
> |:----------------:|:-----------:|:----------:|:----------:|:----------:|:----------:|:----------:|
> | p_gold (±stderr) | 10.29±1.05% | 5.41±0.69% | 4.54±0.73% | 4.04±0.57% | 3.57±0.52% | 3.54±0.57% |
>
> The attention mass drops sharply from 1% to 20% hard proportion, then plateaus across the remaining range. This is consistent with both the pre-softmax logit margins in Figure 5 and the theoretical prediction.
>
> ---
>
> - Q3 Filtering experiment on more models
>
> **TLDR — We extend the experiments to another two models and all datasets, the conclusion is consistent across all settings.**
>
> We extended the filtering experiments to Llama-3.1-8B-Instruct and Qwen2.5-7B-Instruct across all four datasets.
>
> The results (https://anonymous.4open.science/r/89XLms/figure8_filter.pdf) are consistent with our original finding: both filtering strategies yield nearly identical gains over most of the context range, with divergence appearing only when hard proportion drops below the First Drop of Ink threshold. We will include this extended figure in the revision.
>
> ---
>
> - W4: mitigation proposal
>
> We acknowledge that our paper is primarily an analysis contribution, and we believe the negative results are valuable themselves. That said, our findings do point to concrete mitigation directions that we will discuss in the revision, for example, verification before context accumulation or improvement on softmax design.
>
> We consider a full mitigation strategy an important direction for future work, and our analysis provides the necessary foundation for it.
>
> [1] Jin et al. Long-Context LLMs Meet RAG: Overcoming Challenges for Long Inputs in RAG. arXiv 2024.
>
> [2] Du et al. Context Length Alone Hurts LLM Performance Despite Perfect Retrieval. EMNLP 2025

---

> > ### Author Rebuttal · Reviewer_g96m · 2026-04-01
> >
> > All major concerns addressed with new evidence: bootstrap CIs confirm anomalies are noise, post-softmax attention validates the theory, and extended filtering experiments generalize across models and datasets. I raise my score to 5.

---

> > > ### Author Response · Authors · 2026-04-04
> > >
> > > We truly appreciate the reviewer for recognizing our rebuttal. We will update the draft with your suggestions, and thank you again for the valuable advice!

---

### Official Review · Reviewer_iQ4m · 2026-03-13

**Soundness:** 3
**Presentation:** 3
**Significance:** 2
**Originality:** 3
**Overall Recommendation:** 4
**Confidence:** 3

**Summary:**

The problem this paper tackles is that of understanding the impact of misleading but topically relevant documents. To address this, they conduct a set of experiments that vary the effect of the proportion of distracting documents and the "level" of distraction (hard, easy, or random).

**Compliance With Llm Reviewing Policy:**

Affirmed.

**Final Justification:**

The authors have addressed my main concerns and I have increased the overall score.

**Key Questions For Authors:**

1. The paper shows that hard distractors are substantially more harmful than easy/random distractors when inserted at fixed context length, but the filtering results suggest that removing hard versus easy distractors yields similar gains over much of the range. Could the authors clarify how these two findings should be reconciled?


2. Could the authors help clarify whether the main lesson should be to improve retrieval precision upstream, or whether realistic reader-side mitigation can recover a substantial portion of the loss?

3. Would removing random corpus-based distractors behave more like removing easy distractors or more like removing hard distractors? Since random distractors are drawn from the same corpus family as hard distractors, they may provide a more realistic baseline for understanding how distractor-removal-based context shortening interacts with distractor type.

4. The paper hypothesizes that lowering softmax temperature might sharpen attention and reduce the influence of hard distractors, but the experiment shows that this intervention degrades performance. Could the authors elaborate on which part of the underlying assumption fails in practice (e.g., calibration mismatch versus the gold passage not consistently having the highest attention logits)?

**Limitations:**

yes

**Strengths And Weaknesses:**

**Strengths**

The paper highlights an important phenomenon of how long-context QA degrades under misleading context in a strongly nonlinear fashion. This is important to take note of when deploying real RAG systems. The distinction between types of distractors also offer some more nuance. In addition to experiments, it also offers a solid theoretical explanation as well clearly noted takeaways about how the degradation from hard distractors is front-loaded. The experiments and the theory compliment each other well. The paper holds clear diagnostic value for practitioners in heeding the warning that a small amount of semantically plausible noise can do a lot of damage.

**Weaknesses**

1. The relationship between the main “hard distractors hurt more” result and the filtering analysis is not explained clearly enough.

The paper shows that hard distractors are much more harmful than easy/random distractors when added at fixed context length, but the filtering results suggest that removing hard versus easy distractors yields similar gains over much of the range. These findings may not be incompatible, but the paper does not reconcile them clearly, which makes the filtering takeaway harder to interpret.

2. Relation between upstream retrieval precision vs downstream cleanup

The paper’s results suggest that hard distractors are very harmful and difficult to mitigate once included in the context. To better support the practical implication, could the authors compare a small set of upstream retrieval improvements (e.g., reranking or stricter top-k selection to reduce hard distractors) against a small set of downstream cleanup methods (e.g., select-then-answer prompting or evidence filtering)? This would help clarify whether the main lesson should be to improve retrieval precision upstream, or whether realistic reader-side mitigation can recover a substantial portion of the loss.

3. The filtering experiments are diagnostic but not especially realistic as mitigation studies.

The filtering setup assumes oracle knowledge of which passages are hard or easy distractors. This is useful for controlled analysis, but it means the filtering results should be interpreted more as diagnosis than as evidence about deployable filtering pipelines.

4. Some experimental choices leave useful comparisons unexplored.

For example, the filtering analysis compares removing hard versus easy distractors, but not random distractors which is arguably more realistic than easy distractors.

Overall, some of the insights can be more flushed out and the practical takeaways can be strengthened to improve the significance of this work.

---

> ### Author Rebuttal · Authors · 2026-03-26
>
> We thank the reviewer for recognizing the importance of our finding("important to take note of when deploying real RAG systems"), the theoretical contribution ("a solid theoretical explanation"), and the practical value ("clear diagnostic value for practitioners"). Below we address the remaining concerns.
>
> ---
> - The oracle filtering design and reconciliation with main results
>
> **TLDR — Oracle filtering is designed as a diagnostic ceiling; both findings are consistent and explained by the same mechanism.**
>
> We appreciate the reviewer for raising these points. First, the oracle filtering is by design: it represents the ceiling of any downstream filtering approach. Our key finding is that even under this ceiling, filtering hard distractors yields no extra benefit over filtering easy ones — meaning no realistic method can do better, reinforcing our takeaway that upstream precision matters most.
>
> To reconcile with the main result: our main experiments (§3) characterize degradation under fixed context length, while the filtering analysis (§6) explores a more practical scenario where removing documents changes both the hard proportion and the context length simultaneously, creating a confound. Under this setting, we find that removing hard vs. easy distractors yields nearly identical gains over most of the range.
>
> These two findings are explained by the same mechanism. In Table 1, the hard distractor proportion during Filter Hard remains consistently far above the critical threshold (~10%). As our main experiments show, once past this threshold, performance has already plateaued regardless of how many hard distractors are present. Viewed from the removal side: filtering hard distractors only provides extra benefit when nearly all of them are removed; otherwise, the gains come purely from context length reduction.
>
> We will make this connection more explicit in the revised paper.
>
> ---
> - What’s the main lesson from the finding?
>
> **TLDR – Upstream retrieval precision. Reader-side mitigation is largely ineffective unless near-complete removal is achieved.**
>
> Our results suggest that improving upstream retrieval precision is the more critical lesson, rather than putting everything into the context in exchange for higher recall.
>
> Reader-side mitigation: We conduct a small-scale comparison using a select-then-answer pipeline on nq_random (Llama-3.1-8B-Instruct, 128K context): the model first selects the top-30 most relevant passages (3K tokens), then answers. However, the gold passage is included in the model's selection only 32 out of 100 times, and the final accuracy drops from 62.5% to 31%. This indicates the inefficacy of downstream mitigation.
>
> Upstream precision: Meanwhile the filtering experiment in Sec 6 shows that at ~27K, with high precision accuracy reaches 91%. Together, these results clearly favor investing in upstream retrieval precision over downstream cleanup.
>
> ---
> - Removing random distractors as baseline
>
> **TLDR — The suggested experiment already exists in §6.2. "Filter Easy" is a naming issue.**
>
> We want to clarify that the filtering experiment in $\S$6.2 is conducted under nq_random setup, where the weaker distractor type is **Random**, not **Easy**.  “Easy” here was intended to mean the easier typer as opposed to hard distractors. Therefore, the existing experiment already shows the ineffectiveness of post-hoc filtering under realistic corpus-based settings as the reviewer suggested.
>
> We will rename the label to “Filter Non-Hard” or “Random” in the revision to avoid ambiguity. Thank you for pointing this out.
>
>
> ---
> - Why temperature scaling fails
>
> **TLDR — The model’s learned dynamics are coupled with original temperature, so changing it disrupts the entire representation pipeline.**
>
> We confirmed that the gold passage consistently holds a logit margin over distractors and the distribution is indeed sharpened with lower temperature, ruling out the hypothesis that the gold passage lacks dominant attention. Rather, the failure stems from calibration mismatch: all other components (MLP, LayerNorm, etc.) are optimized under the normal temperature 1, and post-hoc adjustment disrupts this coupling. Base on our observation, it leads to degenerate outputs (e.g. repetition, meaningless tokens). This is consistent with [1], who show the Jacobian norm of softmax scales as O($\frac{1}{\tau}$), and [2][3], who confirm temperature scaling is only effective when introduced during training. We will include case analysis in the revision.
>
> [1] Mudarisov. Limitations of Normalization in Attention Mechanism. arXiv 2025.
>
> [2] Ryan. Introducing a Learnable Temperature Value into the Softmax Self-Attention Scores. 2024.
>
> [3] Ram et al. Learning to Focus: Focal Attention for Selective and Scalable Transformers. arXiv 2025.

---

> > ### Author Rebuttal · Reviewer_iQ4m · 2026-04-04
> >
> > Thank you for the detailed response. The rebuttal directly addresses my main concerns. In particular, the authors provide a clear mechanistic explanation reconciling the main results and filtering analysis, and add a concrete downstream filtering experiment that supports their conclusion about the limited effectiveness of reader-side mitigation. They also clarify the role of oracle filtering as a diagnostic ceiling and resolve the ambiguity around the random distractor baseline. Overall, these responses strengthen both the clarity and practical interpretation of the work. Scores will be adjusted accordingly in the final justification.

---

> > > ### Author Response · Authors · 2026-04-04
> > >
> > > We truly appreciate your recognition of our rebuttal and are glad that the concerns have been fully resolved. We will incorporate all discussed improvements in the revision. Thank you again for your time and constructive feedback!

---

### Official Review · Reviewer_sHof · 2026-03-13

**Soundness:** 3
**Presentation:** 3
**Significance:** 2
**Originality:** 2
**Overall Recommendation:** 4
**Confidence:** 3

**Summary:**

This paper studies how the proportion of "hard distractors" (semantically relevant but misleading passages) in long contexts affects LLM performance on QA tasks. The key finding is a nonlinear "first drop of ink" effect: the first subsets of hard distractors causes large performance degradation, while further increases yield diminishing marginal harm. The paper provides a theoretical explanation grounded in the convexity of softmax attention allocation and provide empirical validation on retrieval heads. Controlled experiments across 5 models, 4 datasets, and 6 context lengths demonstrate the robustness of this finding. The paper also shows that filtering benefits primarily come from length reduction rather than composition change, and that temperature scaling is ineffective as mitigation.

**Compliance With Llm Reviewing Policy:**

Affirmed.

**Key Questions For Authors:**

*  The proposed implication of the phenomena suggests prioritizing retrieval precision over recall - this contradicts the common finding that adding more retrieved documents (higher recall) improves performance. Could you provide empirical evidence or elaboration on this claim?
* Aside from quantity of the hard distractor, how does "relevance" of the hard distractor impact the results, and does the position of where the hard distractor is placed impact the performance?

**Limitations:**

Yes

**Strengths And Weaknesses:**

**Strength**
* Understanding how misleading documents affect LLM performance in long contexts is a well-motivated and practically important problem.
* Experiments are comprehensive, covering 4 models with different sizes, multiple QA datasets, and varying context lengths.
* The paper is clearly written.

**Weakness**
* It is somewhat unsurprising to me that even including a small portion of distractor hurts, just like how the model is able to identify relevant passages in the long context. And it is a bit unclear what is the implication for this finding.  In section 7, the paper suggests prioritizing retrieval precision over recall, but this contradicts the common finding that adding more retrieved documents (higher recall) improves performance. Further experiments / elaboration to support the claim should be added.

---

> ### Author Rebuttal · Authors · 2026-03-31
>
> We thank the reviewer for the recognition of the paper’s presentation (“clearly written”) and contribution (“experiments are comprehensive”). Below we clarify the remaining points of confusion.
> - - -
> - It is somewhat unsurprising that even including a small portion of distractor hurts and it’s a bit unclear about the implication.
>
> **TLDR— Our contribution is beyond the simple claim and related with broad long-context applications.**
>
> We appreciate your comment. We would like to clarify our implication and that our contribution is not purely the observation that “distractors hurt”. Rather, our contribution lies in aspects beyond this.
>
> (1) The nonlinearity dynamic is previously unknown. Previous work demonstrates the observation of degradation, yet only as a binary comparison (Lee et al. (2026), Hong et al. (2025)) with/without distractors. Our finding provides the first quantitative analysis of how distractor proportion affects long-context reasoning.
>
> (2) The implication has broad significance. The long-context setting is of increasing importance with long-horizon agent framework, and deep research pipelines, where hard distractors inevitably enter the accumulated context. These systems thus are potentially fragile due to the “First Drop of Ink” effect. Moreover, existing filtering approaches, which were developed and validated at short-context scale, provide only marginal recovery under these long-context scenarios. Together, these findings motivate a shift toward upstream retrieval precision.
>
> (3) The mechanistic analysis offers a different aspect of understanding softmax attention. It offers a further lens into the architecture design.
>
> In summary, we respectfully argue that our finding is far more than “distractors hurt” and the implication is beneficial to broad long-context scenarios.
>
> ---
>
> - “First Drop of Ink” contradicts the common finding that adding more retrieved documents helps.
>
> **TLDR— “More docs help” is not unconditionally true but context-dependent, as supported by previous work. Our work extends it to a long-context scale with empirical and mechanistic analysis.**
>
> We would like to point out that “adding more retrieved documents improves performance” is not unconditionally true, but rather depends on multiple factors as established by prior work.
>
> For **retrieval quality**, Jin et al. (2024) find that when the added documents are “unrelated” negatives, the performance either decreases slightly or even continues to improve. However, when the added documents are hard negatives, the degradation becomes more significant.
>
> For the **context scale**, Xu et al. (2023) show that retrieval is most beneficial within the first 5-10 chunks. Beyond this point, adding more documents tends to hurt the performance (e.g. top-20 underperforms top-5 across different settings in their paper).
>
> Our work further extends by characterizing the nonlinearity of the degradation, disentangling different components (context length, distractor type and proportion) and providing a mechanistic explanation.
>
> **If there are missing details the reviewer had in mind, we would be greatly happy to address them.**
>
> ---
>
> - The impact of relevance and position of hard distractors.
>
> **TLDR— Relevance is captured by the easy/random/hard level already, and the position has no impact on the performance**
> In our work, we divide distractors into three levels of relevance (Hard/Random/Easy). The results in Figure 2 clearly show that hard distractors cause significant degradation, no matter if the rest of the distractors are Easy or Random types.
>
> For the position of hard distractors, we conduct a pilot experiment using Llama-3.1-8B-Instruct under 128K tokens with 200 samples per condition. We vary the proportion of hard distractors from 1% to 80%, and compare performance when these hard distractors are concentrated at the beginning (head) vs. the end (tail) of the context, with the gold document inserted at a random position.
>
> |        Hard %       |  1%  |  20% |  40% |  60% |  80% |  90% |
> |:-------------------:|:----:|:----:|:----:|:----:|:----:|:----:|
> | Head - Tail (Acc %) | -1.5 | -1.0 | +2.5 | -1.0 | +1.0 | -3.0 |
>
> The differences are minor across all proportions with no systematic trend, indicating that the position of hard distractors does not meaningfully affect performance. This also validates our design choice of randomly shuffling document positions (§3.1), ensuring that the observed degradation is driven purely by distractor proportion rather than placement.
>
>
> [1] Lee et al. Lost in the Noise: How Reasoning Models Fail with Contextual Distractors. arXiv 2026.
>
> [2] Hong et al. Context Rot: How Increasing Input Tokens Impacts LLM Performance. Chroma Technical Report, 2025.
>
> [4] Jin et al. Long-Context LLMs Meet RAG: Overcoming Challenges for Long Inputs in RAG. arXiv 2024.
>
> [5] Xu et al. Retrieval Meets Long Context Large Language Models. arXiv 2023.

---

> > ### Author Rebuttal · Reviewer_sHof · 2026-03-31
> >
> > Thank you for the clarification! The added citation of previous evidence that "adding irrelevant doc" hurts is helpful; and that the proportion is more important than location is also an interesting finding to include.
> > I have increased my score to 4. I would encourage the authors to connect the findings to practical downstream utility for developing more capable long-context model / long-horizon agents.

---

> > > ### Author Response · Authors · 2026-04-04
> > >
> > > We truly appreciate the reviewer for recognizing our rebuttal. We will update the draft with your suggestions, and thank you again for the valuable advice!

---

### Decision · Program_Chairs · 2026-04-30

**Decision:**

Accept (regular)

**Comment:**

This paper identifies and addresses the problem in the retrieval-augmented generation setting that "hard distractors," documents that are relevant but are misleading, may degrade the QA performance even if it is just a small proportion of the retrieve documents. They conduct a series of experiments to show the empirical results, as well as a theoretical explanation. While the reviewers pointed out several concerns, most were effectively addressed through the rebuttal. Overall, this is a nice contribution to the ML community in a relevant and important topic.